# Estimating body segment parameters from three-dimensional human body scans

**Pawel Kudzia**[1,2]*, **Erika Jackson**[2], **Genevieve Dumas**[2]

**1** Department of Engineering Science, Simon Fraser University, Burnaby, BC, Canada, **2** Department of Mechanical and Material Engineering, Queen's University, Kingston, ON, Canada

* pawel.kudzia5@gmail.com

## Abstract

Body segment parameters are inputs for a range of applications. Participant-specific estimates of body segment parameters are desirable as this requires fewer prior assumptions and can reduce outcome measurement errors. Commonly used methods for estimating participant-specific body segment parameters are either expensive and out of reach (medical imaging), have many underlying assumptions (geometrical modelling) or are based on a specific subset of a population (regression models). Our objective was to develop a participant-specific 3D scanning and body segmentation method that estimates body segment parameters without any assumptions about the geometry of the body, ethnic background, and gender, is low-cost, fast, and can be readily available. Using a Microsoft Kinect Version 2 camera, we developed a 3D surface scanning protocol that enabled the estimation of participant-specific body segment parameters. To evaluate our system, we performed repeated 3D scans of 21 healthy participants (10 male, 11 female). We used open source tools to segment each body scan into 16 segments (head, torso, abdomen, pelvis, left and right hand, forearm, upper arm, foot, shank and thigh) and wrote custom software to estimate each segment's mass, mass moment of inertia in the three principal orthogonal axes relevant to the center of the segment, longitudinal length, and center of mass. We compared our body segment parameter estimates to those obtained using two comparison methods and found that our system was consistent in estimating total body volume between repeated scans (male p = 0.1194, female p = 0.2240), estimated total body mass without significant differences when compared to our comparison method and a medical scale (male p = 0.8529, female p = 0.6339), and generated consistent and comparable estimates across a range of the body segment parameters of interest. Our work here outlines and provides the code for an inexpensive 3D surface scanning method for estimating a range of participant-specific body segment parameters.

## Introduction

In biomechanics, body segment parameters (BSPs) are required inputs for a range of applications. BSPs include the masses of body segments, the positions of the center of mass of body segments with respect to a segment reference frame, segmental mass moments of inertia with

**Data Availability Statement:** Data are available in the following repository: https://github.com/pkudzia/Paper-BodySegmentParameter.

**Funding:** G.D received funding from NSERC Canada Discovery Grant A6858. The funders had

no role in study design, data collection and analysis, decision to publish, or preparation of the manuscript.

**Competing interests:** The authors have declared that no competing interests exist.

respect to a segment point, and body segment lengths. BSPs can serve as input data for engineering prosthetics [1,2], for ergonomic design [3], and are required for inverse dynamics [4]. Although 'generic' datasets that are composed of measurements from a range of humans can be used for approximating BSPs of people, direct *in vivo* participant-specific estimates provide the highest level of accuracy [5]. Unfortunately, direct approaches to estimate BSPs can be cumbersome and expensive. In this work, we developed and evaluated an easy-to-implement method for indirectly estimating participant-specific BSPs using an inexpensive consumer depth camera.

Participant-specific BSPs can reduce errors associated with biomechanical outcome measures. BSPs estimates are sensitive to the morphology, age, and gender of a person [6,7]. And many biomechanical outcomes measures, such as kinetics, require BSP values to evaluate. Variabilities of +/-5% in BSP estimates can have potentially meaningful effects on the resultant outcomes [8–11]. When comparing different methods used to estimate BSPs, there may also be differences in multiple of the BSPs estimates, further increasing uncertainty in the outcome measures [12]. As some segment BSPs are difficult to estimate (e.g., trunk moments of inertia), using the best available tools to get representative measures should be the goal. The use of participant-specific BSPs estimates is especially important in open-chain or high acceleration motions, such as running and jumping, where there are large body segment accelerations, and in airborne movements, where there are no external forces [13]. Populations that have less available data for making approximations using 'generic' datasets, such as pregnant women [14], amputees [15], and children [6], may also meaningfully benefit from the use of participant-specific BSPs on outcome measure accuracy.

To estimate BSPs requires both density estimates and geometric properties. Medical imaging is the gold standard approach for estimating density. Magnetic resonance imaging [16,17] and computed tomography have been used to estimate *in vivo* BSPs of people by estimating both the geometric and density profiles of the individual body segments [18]. The limitations are that there is exposure to low dose radiation for CT scanning approaches, medical imaging incurs high costs, and for certain populations, these approaches may not be feasible (e.g., pregnant women and children) [6]. Dual-energy x-ray absorptiometry has also shown potential in this field [15] as it is less expensive and faster than the aforementioned approaches. But in general, medical imaging is not readily available and is largely impractical for many laboratory-based experiments seeking inexpensive and minimally involved methodologies.

Indirect methods are then a common approach used for estimating participant-specific BSPs. One well-adopted indirect method is the use of regression models utilizing a person's mass and height as inputs. The convenience of application makes this approach practical, but as BSPs are sensitive to morphology, age, and gender, the use of a regression model on persons who differ from the population that the model is developed on can result in unrepresentative estimates [19]. Some of the most commonly available regression models are derived using data from cadaveric specimens of slender elderly men [20–22] and young adolescents [4]. More so, height and weight as inputs to estimate mass moments of inertia and center of mass positions can be prone to error, where small changes in morphology can create large changes in these parameters, a feature that regression models are not well adapted to capture [23].

Geometrical models coupled with photographic approaches have addressed some of the aforementioned limitations. In geometrical modelling, anthropometric measurements are used to define modifiable shapes that resemble the body segments. Using mathematical principles, the BSPs can be estimated [24,25]. For such an approach, the accuracy of the estimated BSPs most depends on the accuracy of the estimated geometry of the body segments, and less so, but also on, the body segment density values used in the model [26]. What has deterred the widespread use of geometric modelling is the extensive experimental time necessitated for

acquiring anthropometric measurements [25,27]. As such, photographic approaches have been coupled with geometrical models to digitally acquire meaningful anthropometric measurements to then be used as inputs in geometric models [28,29]. One such approach, the elliptical cylinder method, digitizes frontal and sagittal plane photographs to then approximate the anthropometric measures of participants and model the 3D geometry of their body using stacked elliptical cylinders. The elliptical cylinder method has been shown to be accurate to 2.0 ±2.2% in estimating body volume when compared to volume estimates obtained using water submersion (the gold standard in estimating volume by measuring the displacement of water) [30]. However, the digitizing process can be extensive and if the use of a model to estimate the 3D geometry of the body can be mitigated, this would reduce assumptions made about the morphology [31].

3D surface scanning provides an opportunity for acquiring the 3D geometry without using a geometrical model. 3D surface scanning techniques using laser scanning [32], structured light projection [33] and time of flight cameras [34] provide the tools to 3D reconstruct objects, humans, and other animals. 3D surface scanning omits the use of predefined geometrical shapes to estimate the morphology of the body and as a result does not require the use of a 2D photographic method to make anthropometric measures. The Microsoft Kinect Version 1 (Kinect V1, Microsoft Corporation, Redmond, USA) is a low-cost close-range camera that has shown potential for 3D volume estimation [35], for estimating participant-specific anthropometric measurements [36], and in some preliminary work in estimating body segment parameters [37–39]. Volumetric estimations using the Kinect V1 have been reported to have errors of 0.04±2.11%, suggesting greater accuracy than commonly used geometric models [38]. When comparing gold standard medical imaging to those estimated using an array of Kinect V1 cameras (16 cameras in total) a high correlation in total body volume estimation was found ($R^2$ = 0.99) but the Kinect tended to underestimate volume [40,41]. Other 3D cameras have also shown promise in this field of research [19,42–44]. The newest version the Kinect Version 2 (Kinect V2, Microsoft Corporation, Redmond, USA) is more accurate than the Kinect V1 in terms of depth perception and 3D estimation and boasts a higher resolution [34,45,46]. In one recent study, intrinsic dinosaur skulls were 3D scanned using the Kinect V2 where the device was found to perform as well (reported depth resolution of 0.6mm) as industrial-grade laser scanners that cost exponentially more. A consumer depth camera, such as the Microsoft Kinect V2 then presents an opportunity to develop and evaluate an inexpensive approach for estimating participant-specific BSPs while addressing some of the limitations of the aforementioned BSP estimation methods.

In this project, our general goal was to use 3D surface scanning to estimate participant-specific BSPs. As 3D surface scanning has shown to be a promising approach for volumetric and anthropometric measurement estimation, when used in conjunction with density values, it is certainly possible to indirectly estimate the BSPs of humans. In this project, we had three specific aims. The first aim was to develop an experimental approach for collecting participant-specific 3D scans using a readily available consumer depth camera, the Kinect Version 2 (Kinect V2, Microsoft Corporation, Redmond, USA). The second aim was to evaluate a 3D body segmentation procedure for post-processing the 3D scans, and the third aim was to evaluate and assess the BSP estimates obtained using our proposed approach. To accomplish the first aim, we used the Kinect V2 depth camera and performed repeated 3D scans on 21 human participants. To accomplish the second aim, we used MeshLab, an open-source software [47], to segment the body into 16 body segments. To accomplish our third aim, we wrote custom software that evaluated the BSPs of the body segments acquired using our 3D scanning method. We then compared the BSPs obtained using the proposed method to estimates obtained using a camera-based geometrical modelling approach (the elliptical cylinder method

**Table 1. Participants recruited for this study.**

| Participants | Age (Years) | Stature (m) | Mass (kg) | BMI (kg/m²) |
|---|---|---|---|---|
| Males (n = 10) | 23.4±1.7 | 1.8±0.1 | 73.4±4.5 | 22.7±2.1 |
| Females (n = 11) | 22.3±2.4 | 1.7±0.1 | 70.1±8.1 | 23.3±2.3 |

(ECM) [29,48], and to estimates obtained using a regression model derived from a similar population sample.

## Methods

### Participants

We recruited 21 healthy adult participants for our study. The Queen's University ethics board approved the study and we obtained written consent from each participant before participation. We asked each participant to change into form-fitting shorts, a tight t-shirt, and to wear a swim cap on their heads for the 3D scanning [31]. For each participant, we obtained their body mass using a medical scale, measured their stature, asked for their age, and calculated their body mass index (Table 1). We had no participant exclusion criteria.

### Experimental setup

We used an inexpensive depth camera to perform 3D body scans. We mounted the camera, a Kinect Version 2 (Kinect V2, Microsoft Corporation, 2015) onto a generic tripod and connected it via USB 3.0 to our desktop computer. The Kinect V2 has a depth sensor that operates at 30 Hz with a recommend minimum and maximum depth capture distance of 0.5m to 4.5 m respectively [45]. We setup the computer to operate on Windows 8.1 running an Nvidia GTX 970 graphics processing unit, a Z97 Gaming 3 intel motherboard, with 16GB of RAM. We used a 30-foot active USB 3.0 extension cord (SuperSpeed USB 3.0 Active Extension) to extend the reach of the device from the workspace computer ensuring that the camera could freely be moved around the room. We constructed a transparent platform for the participants to stand on during their 3D scans (Fig 1). This platform served to maximize the viewing of the participant's feet while minimizing the view of the floor surface. For 3D data acquisition, we used Microsoft 3D Builder (Microsoft Corporation, 2015), free software that integrates with the Kinect V2 and saves the 3D point cloud data for viewing in a user interface.

### Segmentation boundaries

We placed anatomical landmarks on each participant to be used as body segmentation boundaries in post-processing. A trained operator identified landmarks in the frontal and sagittal planes and placed body markers following a set of common guidelines (see S1 File). To minimize volume artifacts, we used flat generic sticker markers 2cm in diameter. After we identified the anatomical landmarks, to allow for clear and repeatable segmentation boundaries, we wrapped a band of non-reflective tape around each limb such that the frontal and sagittal landmarks connected. This approach worked well in our pilot experiments and we used it here to minimize variability in limb segmentation between repeated scans by providing reproducible and clear endpoint boundaries for each body segment.

### 3D scanning

Participants stood on the transparent platform while we revolved the Kinect around them to capture a 3D scan of their bodies. We asked each participant to stand with their feet separated

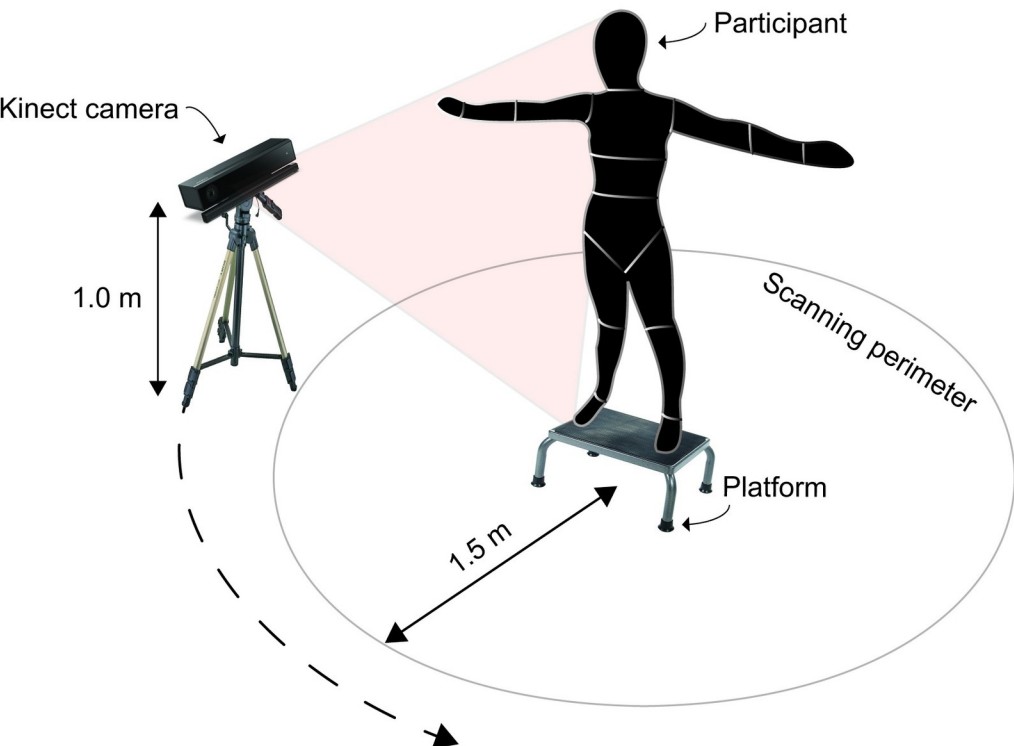

**Fig 1. Experimental setup used for collecting 3D body scans.** We manually revolved the Kinect V2 camera around the participant following the scanning perimeter. Participants stood with their arms abducted on a transparent platform.

and arms abducted at approximately 90˚. We found that this scanning posture allowed for maximal visibility underneath the arms and in between the thighs, to minimize any scanning errors resulting in skin folding when arms are together or thighs touching. Using a verbal cue "*we are now starting the 3D scan*", we vocalized to the participant the starting of the scan. At this moment the computer operator initiated the acquisition software while a second operator began to manually revolve the Kinect tripod around the participant. The operator revolved the camera following an outlined path around the participant, holding the tripod at a vertical height of 1m (Fig 1). After a full revolution of 360˚ around the participant, the operator lifted the tripod to a higher position of ~2m by extending the legs of the tripod rapidly and continued walking around the participant until a second revolution was complete. In pilot experiments, we found that by raising the Kinect tripod in the second revolution a more complete view of the superior aspects of the upper body segments could be captured resulting in visually more complete scans. As a result, each complete participant scan that we used for analysis consisted of these two aforementioned revolutions. We asked participants to remain as still as possible during the scanning process and withhold from breathing deeply to minimize any scanning artifacts [31]. Once the scan was finished, we informed the participants they could relax. We immediately reviewed the quality of the scan on the computer and considered it to meet our inclusion criteria if no visual obstructions or evident volume deformations were visible. These deformations could occur if the participants had moved or shifted their position during the scan (e.g., swaying, or moving arms). For each participant, we sought to collect 3 scans that met these inclusion criteria. Each body scan took ~30 seconds and each participant required at least 5 repetitions until 3 scans meeting our inclusion criteria were collected.

## Data analysis

We exported all 3D scans and processed them for analysis. From Microsoft 3D Builder, we exported each scan as a polygon file format (.ply) to preserve texture (i.e., colour of the pixels) for segmentation. We then imported each scan into MeshLab (64-bit v.1.3.4) [47] an open-source software we used for processing and editing the 3D data. In MeshLab, we took the extraneous point cloud data, such as the floor and ceiling, and deleted it manually using the graphical user interface features. Next, we aligned each scan such that the global coordinate system in MeshLab best aligned with the anatomical planes of the body (Fig 2). This approach helped us to ensure that segmentation along the frontal plane of the body was perpendicular to the plane itself allowing us to produce repeatable segmentations. Following the aforementioned alignment procedure, we segmented each scan manually into a 16-segment model using the segmentation boundaries as the guidelines (Fig 3). The body segments in the segmentation include the left and right hand, forearm, upper arm, thigh, shank, and foot as well as the head, torso, abdomen and pelvis. Finally, we saved each body segment as a Stanford triangle format (.STL) and then imported this point cloud data into MATLAB (MathWorks, 2019a) for further analysis.

We wrote custom-written scripts to evaluate the BSPs of each body segment. We wrote MATLAB scripts for evaluating the following outcome parameters: body segment volume,

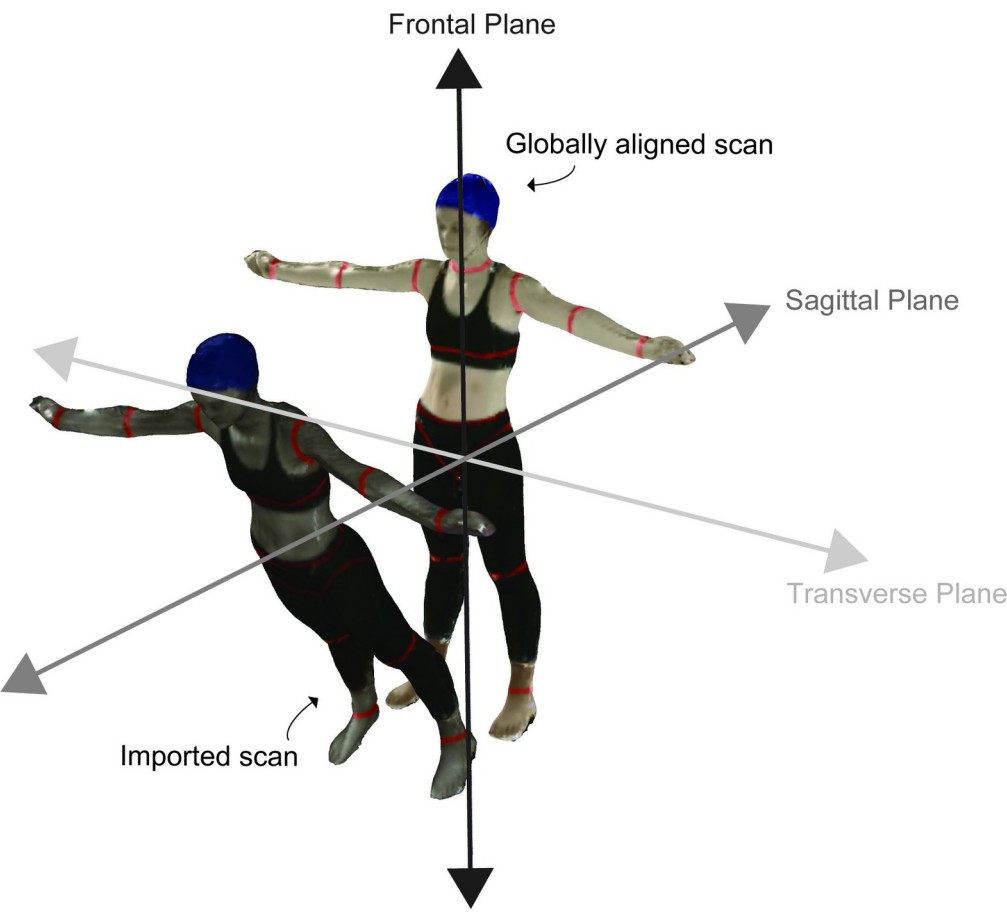

**Fig 2. We aligned imported 3D body scans such that body segment segmentation along the frontal plane was perpendicular to this plane.** We aligned the imported 3D scans to the MeshLab global coordinate system as shown.

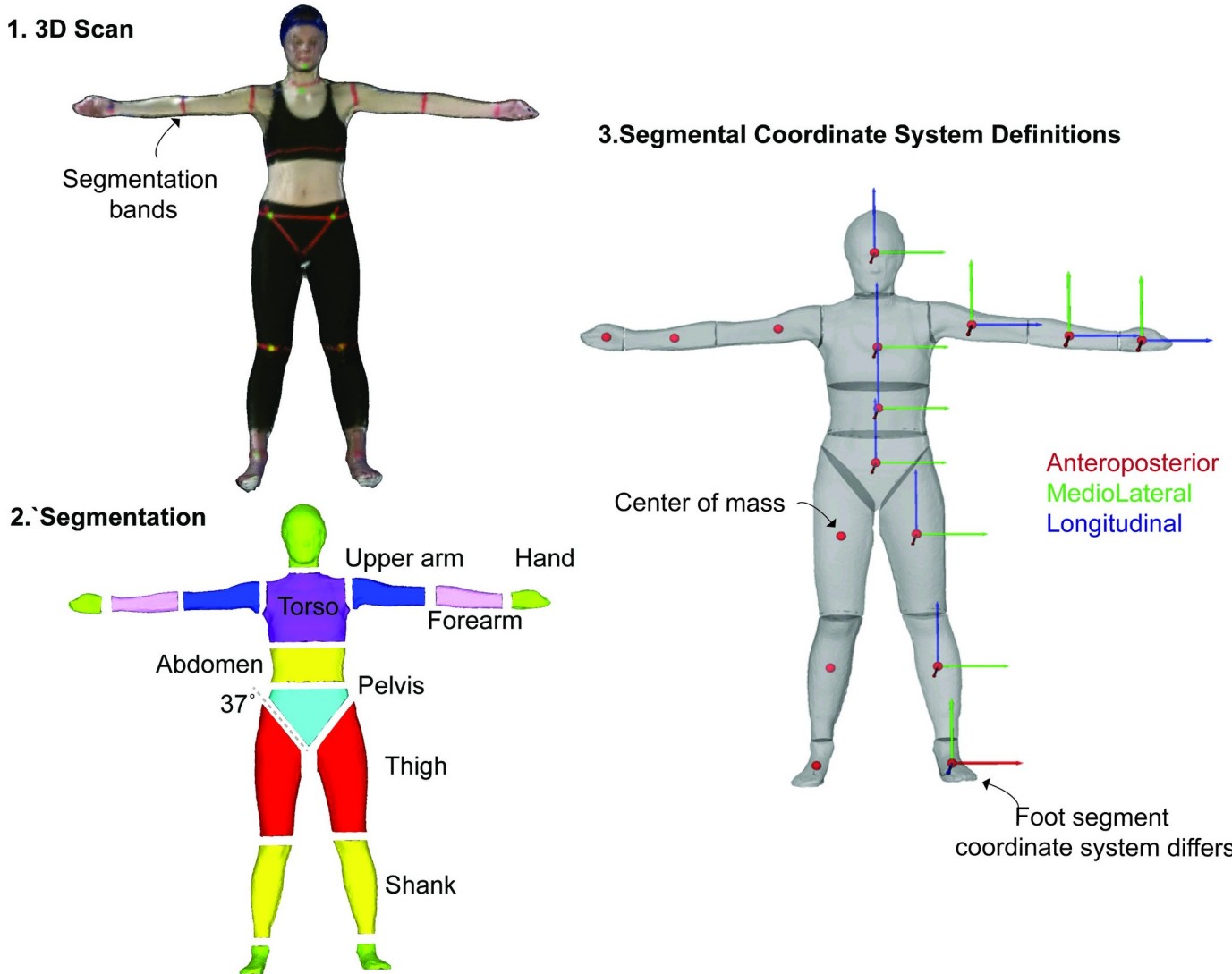

**Fig 3. From 3D scan to BSP output estimations. 1.** A representative 3D scan in MeshLab. **2.** We segmented each 3D scan into 16 individual body segments. **3.** Segmental coordinate system definitions for each body segment showing anteroposterior (red), mediolateral (green), longitudinal (blue) axes. Here the red dot is the center of mass. We aligned the foot segment coordinate system such that the longitudinal axe was along the long length of the foot.

segment mass, the mass moment of inertia tensor about the center of mass of each segment, the longitudinal segment length, and the center of mass position of each segment. We calculated the total volume of each segment as the total encapsulation space of the point cloud data [49]. We then determined the mass of each segment by multiplying its volume by a corresponding uniform density value commonly used in literature (see S2 File for values). The density inputs we used were derived from cadavers of an older population [22] and from CT trunk estimates of a more closely related population [18]. We then calculated the total body mass of each participant by adding up the mass of each of the 16 body segments. We determined the geometrical 3D center of the encapsulation space of each body segment and used this as the estimated center of mass [49]. We took this approach as opposed to taking the average of all the points that describe each segment, as we found that the latter may result in small errors in the center of mass approximation [31]. We calculated the segment mass moments of inertia

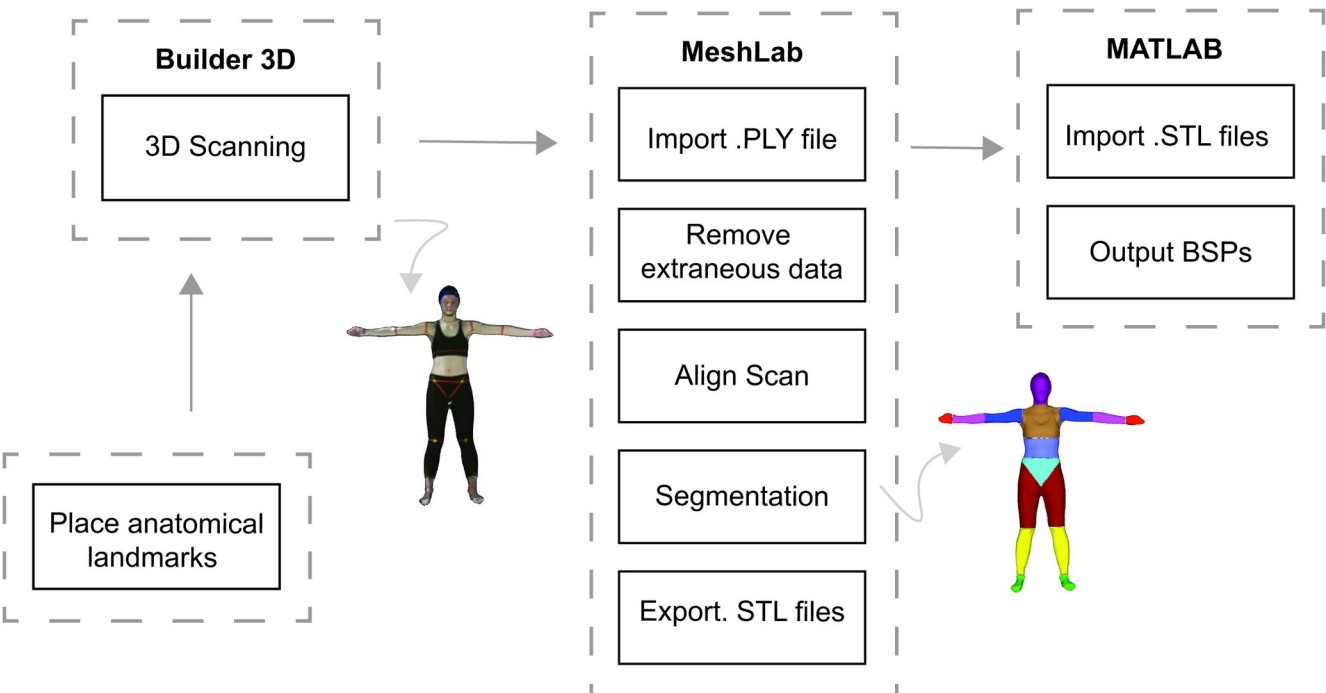

**Fig 4. Overview of our workflow.** After placing anatomical landmarks on the participant, we used a Microsoft Kinect V2 and Microsoft Builder 3D to collect repeated 3D body scans. We exported the acquired scans into MeshLab for removing extraneous data, globally aligning, and segmenting each scan into 16 body segments. We then wrote custom MATLAB scripts to estimate the BSPs of each body segment.

tensor using the center of mass as the reference frame. Although we estimated the full inertial tensor, our primary interests were in comparing principal axes of this tensor along the three orthogonal axes consisting of the anteroposterior ($I_{ap}$), medio-lateral ($I_{ml}$) and longitudinal ($I_{long}$) axes of each segment (for coordinate system definitions refer to Fig 3). We then determined the longitudinal segmental length by projecting a vector from the center of mass position along the longitudinal axis of the segment until this line intersected a segment endpoint on each end of the segment [31]. The total length was then evaluated as the distance from one endpoint to the other endpoint. We used the estimated longitudinal length of the segment to evaluate the position of the center of mass in terms of its distance from the proximal endpoint of the segment (pCOM) (see S3 File for visual illustrations of longitudinal segment lengths and proximal and distal endpoints of each body segment). We show the high-level overview of our workflow for this experiment in Fig 4.

## Comparison methods

We sought to compare BSPs evaluated using the proposed method to the elliptical cylinder method [30,48]. In brief, to evaluate BSPs using the elliptical cylinder method, we placed two digital cameras (Fujifilm Finepix AX 600) on tripods. Each camera was 5m from the participant at a height of 1m. One camera was oriented to capture the frontal plane and another the sagittal plane. We hung meter sticks from the ceiling to serve as a tool for image calibration in post-processing. The meter sticks were hung such that a horizontal and a vertical one was visible in each of the frontal and sagittal planes of the photographic images (see S4 File for illustration and full details). We asked each participant to stand on an inclined platform that is commonly used in this photographic method to capture the feet [30,48]. We asked participants

to stand with their arms at their chest fully extended with their hands supinated and shoulders flexed (see ref [31] for more details). Once the participant was ready, we asked them to stand still while we captured the two images simultaneously, one from each camera. We imported the images into the Slicer Project [50] software that has been adapted for the elliptical cylinder method guidelines for estimating body segment parameters using the imported photographic images. A single trained operator then digitized all of the photographs using this software. Finally, we adjusted the coordinate systems for the output BSPs to best align with those evaluated using our proposed 3D scanning method for comparisons (again refer to Fig 3 for specific axes definitions).

We also compared our participant-specific data to estimates obtained using a regression model. We used the regression model developed using data from a similar population pool [4]. The regression model by Zatsiorsky et al. was developed using a protocol that performed medical imaging on 100 adult Caucasian males (age: 23.8±6.2 years, height: 1.74±0.06 m, mass: 73 ±9.1 kg, body mass index: 24) and 15 adult Caucasian females (age: 19.0±4.0 years, height: 1.74 ±0.03 m, mass: 61.9± 7.3 kg, body mass index: 20.5) (regression equations in S2 File). To estimate the body segment parameters of our participants, we used this regression model by using each participant's measured height and weight as inputs to the model. The similar population that these equations are derived from when compared to the participants in this study makes them a reasonable choice for use for comparison. When necessary, we adjusted the BSP axes such that the output BSPs most closely aligned with those used in our proposed method (again refer to Fig 3 for axes definitions).

## Scanning of a cylindrical object

To get an approximation of the relative accuracy of the device and verify that our approach was working as intended we scanned a cylindrical beam 25 times using a modified version of our 3D scanning protocol outlined above (for beam values and calculations see S5 File). We determined the mathematical geometrical expressions for the inertial parameters of the beam including total volume, longitudinal length, the proximal center of mass position, and the mass moments of inertia in the three orthogonal principal axes of the beam. We found that the 3D scanning estimates were within a reasonable range when compared to the theoretical predictions. For example, when comparing the total volume of the beam using the mathematical expression (4964 cm$^3$) to our methods (5173±204cm$^3$) our 3D scanning method overestimated volume on average by +4.2%. When comparing the longitudinal length (expression: 94.1cm; our methods 93.6±20.7cm) and pCOM (expression: 50%; our methods: 49.8±0.9%) our approach on average underestimated length by approximately -0.6% and pCOM by -0.3% respectively. The orthogonal mass moments of inertia approximated using our approach differed on average by less than +1% (anteroposterior axes $I_{ap}$ = expression: 3676 kg/cm$^2$; our methods 3707±205 kg/cm$^2$; mediolateral axes $I_{ml}$ = expression: 3676 kg/cm$^2$; our methods 3702±205 kg/cm$^2$) with the largest difference observed for the longitudinal axes of the beam of +7.8% (longitudinal axes $I_{long}$ = expression: 42 kg/cm$^2$; our methods 45.3±4.6 kg/cm$^2$). This experiment gave us confidence that the outputs we found were within a reasonable range of what we expected to find.

## Experimental outcomes

We assessed the outcome measures here using our proposed method on males and females separately.

**Total body volume.** To determine the reliability in collecting 3D body scans we used each participant's total body volume obtained from each of the 3 scans which we refer to as Scan A,

Scan B, and Scan C. We used a 1-way repeated-measures ANOVA to test for differences in mean total body volume between repeated scans. We also calculated the 2-way mixed-effects intraclass correlation coefficients (ICC) to provide estimates in the consistency of the estimated outcomes. Following recommended guidelines we considered ICC (2,1) = < 0.5 as poor, 0.50–0.75 as moderate, 0.75–0.9 as good, and >0.9 as excellent [51]. We calculated the coefficients of variations (CV = mean/standard deviation x 100) of body volume estimation to express a measure of normalized variability between repeated scans. We considered coefficients of variations >15% as not acceptable, 15–10% as acceptable, 10–5% as good, and <5% as very good. As we could not find a consensus for acceptable values for coefficients of variation and arbitrarily determined acceptable values widely range between fields of research [52]. We therefore based our considerations using a commonly reported cut-off value of 15%.

**Total body mass.** We evaluated total body mass estimates obtained using our proposed method, to those determined using the elliptical cylinder method, and to the medical scale (our gold standard mass estimate). We used a 1-way ANOVA to test for differences between the mean predicted body mass between the methods. We also evaluated the total body mass agreements between our 3D scanning estimates and the medical scale using a Bland-Altman approach [53]. Here we found the limits of agreement by comparing the differences between the two methods and report these limits along with any found bias. A positive bias is an indication that the 3D scanning approach overestimates mass whereas a negative bias is an indication of underestimating mass. To minimize possible effects and assumptions associated with our use different estimated density values for each body segment to calculate total body mass, we also evaluated the limits of agreement and bias using a standard uniform density value of 1000 kg/m$^3$ (density of water) across all body segments.

**3D segmentation reliability.** We performed the 3D body segmentation following our proposed protocol and evaluated BSPs for each body segment. The BSPs of interest were body segment masses, mass moment of inertia estimates in the anteroposterior ($I_{ap}$), mediolateral ($I_{ml}$), and longitudinal ($I_{long}$) axes, segmental longitudinal lengths, and segmental centers of mass. To evaluate the reliability of the estimates we obtained from these measurements we calculated the coefficients of variations (CV) across the 3 separate segmentations of each body segment for each BSPs of interest. We then compared the coefficients of variations between segments across the three scans and used a 1-way repeated-measures ANOVA to test for differences between the estimated BSPs across segmentations. Following the same approach as with the total body volume, here we also calculated the 2-way mixed-effects intraclass correlation coefficients (ICC) to provide estimates in the consistency of the estimated BSPs across segmentations. We compared only the right side of the body for all segments that had both a left and right side.

**Body segment parameter estimates.** We compared the values of the BSP estimates found using our proposed method, against the BSPs estimated obtained using the ECM approach, and to values obtained using the comparison regression analysis for each body segment (right arm, right forearm, right hand, right thigh, right shank, right foot, head torso, abdomen and pelvis). We used the mean values determined from the 3 scans when using our proposed approach and compared this against the values obtained from the ECM approach, and the regression estimates. We compared the mean segmental mass estimates, and the mass moment of inertia estimates in the anteroposterior ($I_{ap}$), mediolateral ($I_{ml}$), and longitudinal ($I_{long}$) axes. For comparison of the estimated longitudinal segment lengths and proximal positions of the segmental centers of mass the regression equations did not provide equations for length. Therefore, we used mean values from Zatsiorsky and Seylanov for comparisons [4]. We used a 1-way ANOVA to test for differences between estimates obtained using our proposed method, the ECM method, and the regression analysis for each BSP outcome.

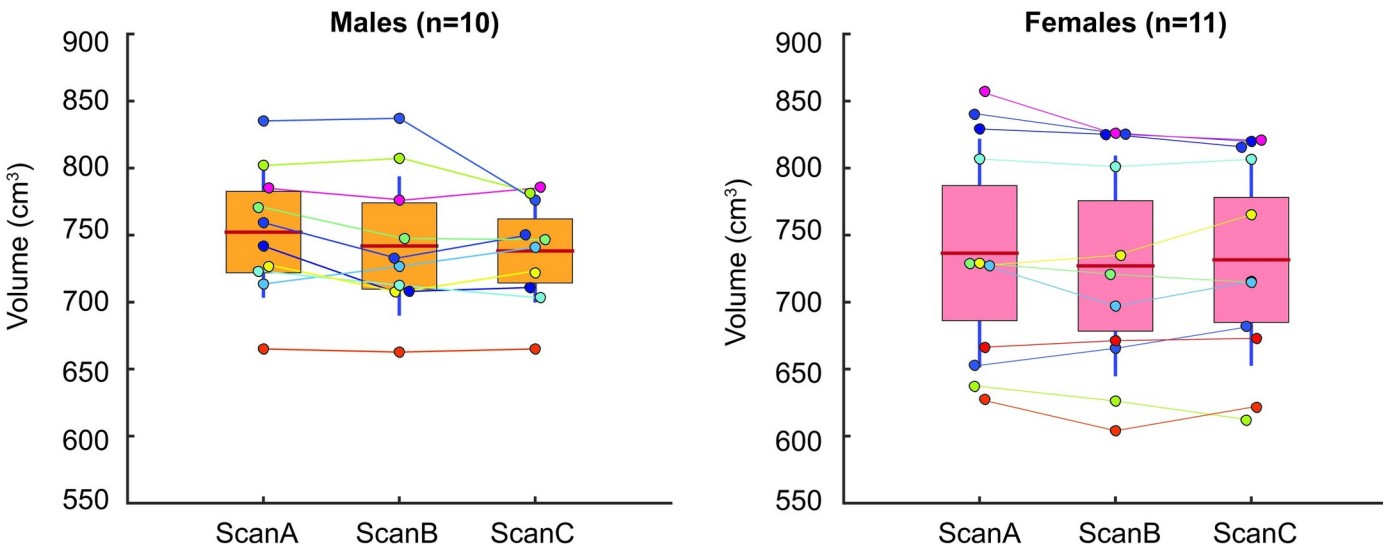

**Fig 5. Total body volume estimated with our proposed 3D method for both males and females.** Comparison volumes between the three scans (Scan A-C) are shown for each participant (coloured dot). The box is the 95% confidence interval around the mean (red line) and SD about the mean is shown (vertical blue line).

We performed all statistical tests using the MATLAB statistical toolbox. Unless otherwise indicated, we report inter-participant means and standard deviations (SD). For the repeated-measures ANOVAs we performed a Mauchly test for sphericity. If the sphericity assumption was violated, we used epsilon adjustments factors. In the event of a statistically significant main effect, we performed post-hoc pairwise comparisons with Bonferroni corrections [54]. We set the level of significance at 0.05 for all statistical analyses.

## Results

### High reliability in total body volume between repeated 3D scans

The total encapsulation volume between the three repeated 3D scans did not differ significantly for both males ($p = 0.1194$) and females ($p = 0.2240$) (Fig 5) (for data refer to S6 File). The total body volume variability exhibited between repeated scans was similar for both males (SD $13.4 \pm 8.9$ cm$^2$, CV $1.8 \pm 1.1\%$) and females (S.D. $11.5 \pm 5.9$ cm$^2$, CV $1.6 \pm 0.8\%$) with CV $<3\%$ (very good). We found high ICC estimates of ICC $(2,1) = 0.96$ for the males and ICC $= (2,1) = 0.99$ for the females corresponding to excellent total body volume repeatability between scans.

### Reliable estimates of total body mass with 3D scans

We found that for males there were no significant differences ($p = 0.8529$) between the average predicted total body mass estimated using the proposed method ($74.5 \pm 4.5$ kg), the ECM method ($73.8 \pm 4.7$ kg), and medical scale ($73.4 \pm 4.5$ kg) (Fig 6). For female participants, we found no significant differences ($p = 0.6339$) between the average predicted total body mass estimated using the proposed method ($73.2 \pm 8.0$ kg), the ECM method ($70.7 \pm 8.1$ kg), and the medical scale ($70.4 \pm 8.1$ kg) (Fig 6).

When comparing the total body mass predictions from all of our 3D scans to the medical scale mass for males, we found limits of agreement from 2.8 to -5.0kg (+1.96 SD to -1.96S SD) with a mean difference (bias) of -1.1kg (-1.5%). For females, we found limits of agreement of 0.64 to -6.9kg (+1.96 SD to -1.96 SD) with a mean difference (bias) of -3.1 kg (-4.4%)

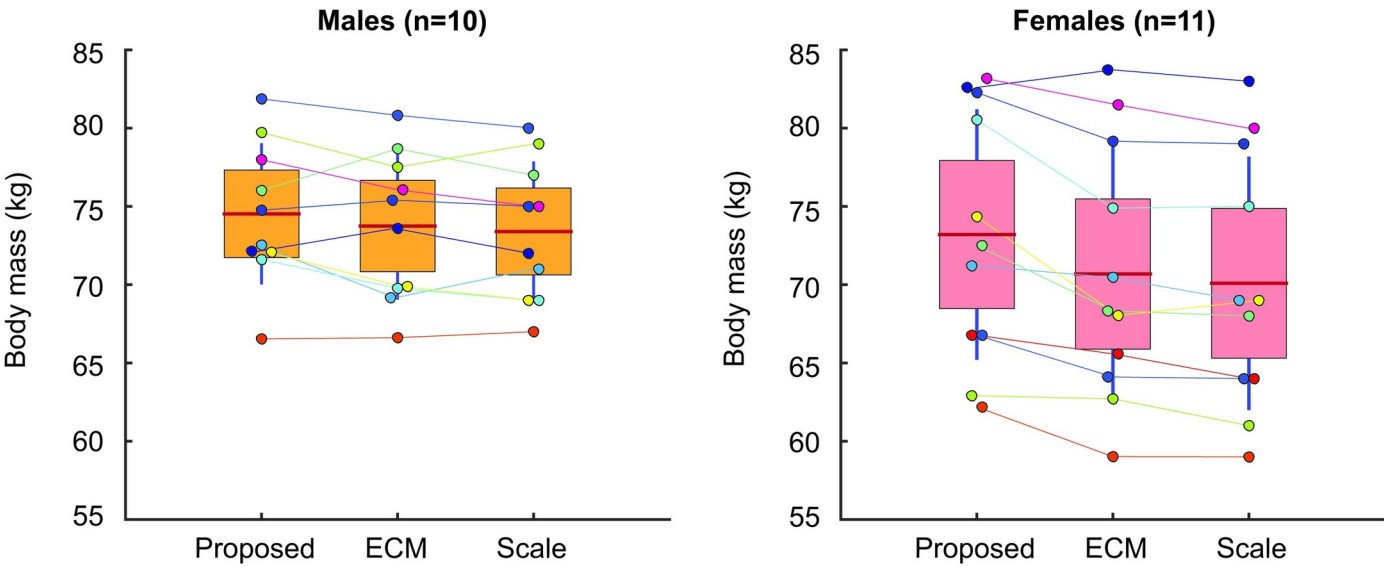

**Fig 6. Body mass (kg) estimated with our proposed 3D method, the ECM method, and the medical scale for both males and females.** Each participant (coloured dot). The box is the 95% confidence interval around the mean (red line) and SD about the mean is shown (vertical blue line).

($p<0.001$). Performing this analysis using a constant density of 1000 kg/m$^3$ across all body segments resulted in results that led to the same conclusions.

### 3D body segment segmentation was most reliable for larger body segments

The body segmentation of each scan (i.e., 16 body segments per scan) took ~30–40 minutes from importing to BSP estimations. The total time for post-processing decreased as we became more proficient in applying the protocol. The main fraction of this time was in the manual nature of the segmentations in MeshLab. The estimated BSPs were predominately repeatable between the segmentation of the three scans (Table 2). We found significant differences ($p<0.05$) between CV's for some of the repeated measures of each body segment parameter outputs (as indicated by an asterisk * in Table 2) with most of these differences observed in the

**Table 2. The mean coefficients of variations (%) for repeated body segmentation of repeated 3D scans.**

| Segment | Male Coefficients of variation (%) | | | | | | Female Coefficients of variation (%) | | | | | |
|---|---|---|---|---|---|---|---|---|---|---|---|---|
| | Volume | $I_{ap}$ | $I_{ml}$ | $I_{long}$ | Length | pCOM | Volume | $I_{ap}$ | $I_{ml}$ | $I_{long}$ | Length | pCOM |
| Head | 3.0 | 7.0 | 6.3* | 4.0 | 2.7* | 0.8 | 4.0 | 8.7 | 7.4 | 5.8 | 2.9 | 2.0 |
| Torso | 2.9 | 5.1 | 6.2 | 4.0 | 3.8 | 1.7 | 3.6 | 6.8 | 7.4 | 4.1 | 4.1 | 1.6 |
| Ab | 2.9 | 4.9 | 6.1 | 4.0 | 2.8 | 0.8 | 2.8 | 4.0 | 5.4* | 3.2 | 2.9* | 2.0 |
| Pelvis | 4.4 | 8.2 | 6.6 | 6.0 | 4.7* | 2.4 | 7.7 | 14.3 | 11.3 | 10.5 | 5.9 | 3.0 |
| Thigh | 4.6 | 8.1* | 8.1* | 8.0 | 2.4* | 1.2 | 3.3 | 7.1 | 7.5 | 5.2 | 2.7 | 1.4 |
| Shank | 5.7 | 6.9 | 7.0 | 11.0 | 1.3 | 0.8* | 3.5 | 5.1 | 5.3 | 6.1 | 1.1 | 1.4 |
| Foot | 13.2 | 19.4 | 19.1 | 25.0 | 5.1 | 5.1* | 11.0* | 16.1* | 15.2* | 19.3 | 3.9 | 4.7 |
| Arm | 9.5* | 19.8 | 20.4 | 18.0 | 4.3* | 2.4 | 7.4 | 11.7 | 12.5 | 14.4 | 3.0 | 3.5 |
| Forearm | 7.1 | 12.8 | 12.9 | 10.0 | 3.3 | 1.8 | 13.3 | 19.9 | 21.1 | 19.2 | 3.7 | 2.6 |
| Hand | 16.6* | 30* | 30.7* | 23.7* | 8.9* | 4.9 | 20.2* | 30.3* | 30.9* | 29.8* | 9.2 | 3.9 |

BSPs were evaluated for each segmentation for both male and female participants. Here an asterisk (*) indicates a $p<0.05$ for the repeated-measures comparisons.

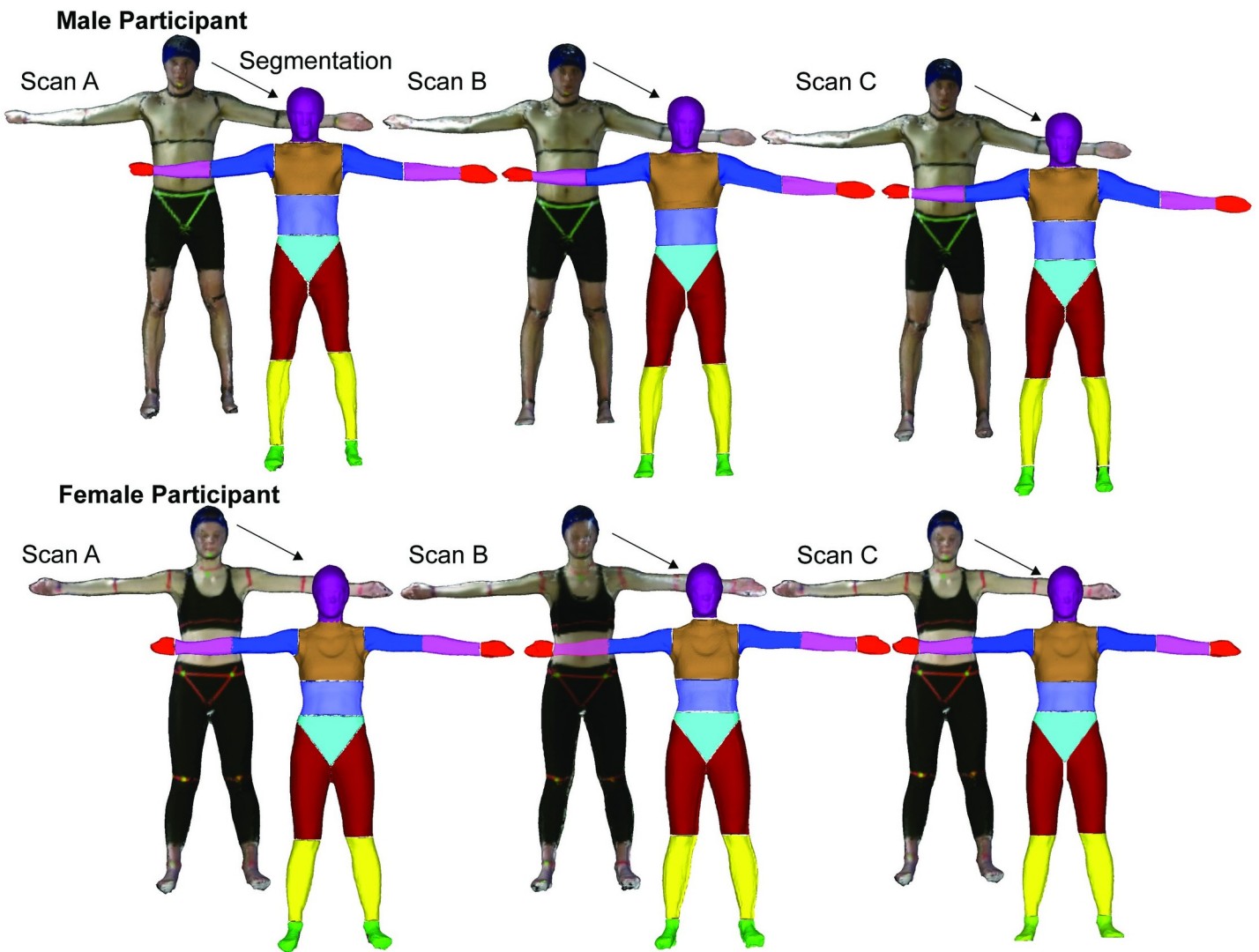

**Fig 7.** Three different scans (Scan A, Scan B, Scan C) and three individual body segmentations from a representative male and female participant are shown. The colouring used for the segmented scan is meant to show clear segmentation borders between adjacent body segments.

smallest segments such as the hand and foot. These small segments also had the largest coefficients of variations for both the male and females and in some instances the CV values were above our accepted cut-off (>15%) suggesting higher variability for these segments. The larger body segments, such as the torso, had the lowest coefficients of variations throughout all of the body segment parameters estimates. A visual representation of the segmentation for one representative male and one female participant for three repeated scans and segmentations is presented in Fig 7. In most instances, we found high ICC estimates for all of the evaluated BSPs across repeated segmentations corresponding to excellent reliability (ICC (2,1)>0.9) (see S6 File for full table). We did however find that for a few BSPs and for certain body segments the ICC estimates were poor (ICC (2,1) <0.5), suggesting poor repeatability. This was observed for both males and females and in most cases for the smaller distal body segments specifically the foot, hand, and arm.

**Table 3. Body segment mass estimates (mean±SD in kg) obtained using the proposed method, the ECM approach, and regression modelling for both male and female participants.**

| Segment | Male Segment Mass (kg) | | | | | Female Segment Mass (kg) | | | | |
|---|---|---|---|---|---|---|---|---|---|---|
| | Proposed | | ECM | | Regression | Proposed | | ECM | | Regression |
| Head | 5.6±0.4 | ● | 5.7±0.3 | △ | 5.1±0.1 | 5.3±0.4 | | 5.4±0.6 | △ | 4.9±0.1 |
| Up-Trunk | 11.5±1.0 | ○ | 10.1±0.8 | △ | 11.4±0.8 | 12.0±2.0 | | 10.4±2.0 | | 10.5±1.9 |
| Abdomen | 11.0±1.6 | | 10.5±1.7 | | 11.7±1.0 | 7.0±1.2 | ● | 6.4±0.9 | △ | 9.1±0.7 |
| Pelvis | 9.2±0.8 | ● | 8.9±0.6 | | 8.4±0.6 | 8.9±1.4 | | 8.5±1.6 | | 8.5±1.2 |
| Thigh | 10.4±0.8 | | 10.4±1.0 | | 10.5±0.7 | 12.0±1.4 | | 11.7±1.5 | | 10.7±1.3 |
| Shank | 3.5±0.4 | | 3.6±0.5 | | 3.2±0.2 | 3.9±0.5 | ● | 3.8±0.5 | △ | 2.9±0.2 |
| Foot | 1.1±0.2 | | 1.1±0.2 | | 1.0±0.1 | 1.1±0.2 | ● | 1.0±0.1 | △ | 1.5±0.1 |
| Arm | 2.1±0.2 | | 2.3±0.3 | △ | 2.0±0.1 | 1.9±0.2 | | 2.0±0.3 | △ | 1.7±0.1 |
| Forearm | 1.1±0.1 | | 1.2±0.2 | | 1.2±0.1 | 0.9±0.2 | | 1.0±0.1 | | 1.0±0.1 |
| Hand | 0.5±0.1 | ○ | 0.7±0.1 | △ | 0.5±0.0 | 0.4±0.1 | ○ | 0.5±0.1 | △ | 0.4±0.0 |

Significant differences (p<0.05) found in post-hoc analysis are reported as symbols. Where '○' indicates a significant between the proposed and the ECM method, '●' indicates a significant difference between the proposed and the regression modelling, and '△' indicates a significant difference between ECM and regression modelling.

## Estimated body mass and mass moments of inertia compared well to comparison methods

We found that the proposed method reliably estimated participant-specific BSPs with estimates that were generally comparable to those determined using the ECM and the regression modelling approaches. However, we did find differences between methods and across body segments. The mean estimated body segment masses (Table 3) were close in magnitude when compared against the comparison methods across all of the body segments with some clear differences across several segments (e.g., Head, and Up-Trunk for males and Shank and Foot for females). We also found some differences in the estimated mass moments of inertia in the anteroposterior ($I_{apl}$), mediolateral ($I_{frontal}$), and longitudinal ($I_{lg}$) principal axes from each segments center of mass reference frames again for similar body segments as mentioned above (Table 4).

## Differences in longitudinal length and center of mass estimates across methods

We compared estimates for the longitudinal length (Table 5) and center of mass position from the proximal endpoints (pCOM) (Table 6) using our approach, the ECM approach, and from average values from literature. Although our method capability estimated length and center of mass within a similar range to the other two methods, our comparison with the ECM approach revealed significant differences across many of the segments for both male and female participants. As the longitudinal lengths were used to determine the pCOM estimates, differences seen in longitudinal lengths estimates are also evident in the pCOM estimates.

## Discussion

We evaluated an inexpensive 3D surface scanning approach for estimating participant-specific BSPs. We used a readily available consumer depth camera, the Kinect V2 to collect repeated 3D body scans of 21 participants. Interaction with the participant for acquiring the 3D scan took around 20 minutes (broken down to between 15–20 minutes for landmarking, and 30 seconds per scan). The post-processing from importing the 3D scan to outputted BSPs took

**Table 4. Moment of inertia estimates (mean±SD in kg cm$^2$) obtained using the proposed method, the ECM approach, and regression modelling for both male and female participants.**

| | Male I$_{ap}$ (kg cm$^2$) | | | | | Female I$_{ap}$ (kg cm$^2$) | | | | |
|---|---|---|---|---|---|---|---|---|---|---|
| Segment | Proposed | | ECM | | Regression | Proposed | | ECM | | Regression |
| Head | 289±36 | ○ | 248±35 | | 280±12 | 261±38 | ○\|● | 197±34 | | 226±1 |
| Up-Trunk | 1839±258 | ○ | 1237±175 | △ | 1708±158 | 1874±517 | ○ | 1355±456 | | 1434±376 |
| Abdomen | 1037±244 | | 962±276 | △ | 1234±174 | 503±134 | ● | 438±96 | △ | 732±68 |
| Pelvis | 635±100 | | 650±104 | | 700±82 | 620±149 | | 641±262 | | 709±144 |
| Thigh | 1682±183 | ● | 1866±288 | | 2104±209 | 2028±431 | | 2086±459 | | 2082±477 |
| Shank | 394±67 | | 454±84 | | 420±52 | 392±90 | | 457±116 | | 354±73 |
| Foot | 48±14 | | 49±9 | | 47±5 | 43±17 | | 41±11 | | 39±8 |
| Arm | 156±35 | ○ | 197±36 | △ | 135±13 | 138±32 | | 146±32 | | 126±15 |
| Forearm | 55±12 | ○\|● | 70±12 | | 67±5 | 43±15 | | 49±12 | | 54±12 |
| Hand | 8±5 | ○\|● | 17±4 | | 14±1 | 6±2 | ○\|● | 11±3 | | 9±1 |
| | I$_{ml}$ (kg cm$^2$) | | | | | I$_{ml}$ (kg cm$^2$) | | | | |
| Head | 331±4 | ○\|● | 278±39 | | 303±14 | 323±5 | ○\|● | 255±46 | | 259±11 |
| Up-Trunk | 1057±170 | ○\|● | 825±115 | | 684±80 | 1319±385 | ○\|● | 977±318 | △ | 648±193 |
| Abdomen | 754±194 | | 683±187 | | 790±116 | 335±108 | ● | 290±74 | △ | 540±65 |
| Pelvis | 688±112 | ● | 680±107 | △ | 548±63 | 645±178 | | 647±180 | | 516±101 |
| Thigh | 1756±189 | ● | 1959±291 | | 2107±213 | 2126±449 | | 2213±467 | | 2077±449 |
| Shank | 397±68 | | 463±87 | | 406±53 | 392±93 | | 430±170 | | 350±71 |
| Foot | 45±11 | | 48±9 | | 43±5 | 40±15 | | 38±15 | | 33±6 |
| Arm | 148±37 | ○ | 205±37 | △ | 121±12 | 129±28 | | 132±63 | | 100±23 |
| Forearm | 55±13 | ○ | 68±11 | | 62±5 | 42±15 | | 44±16 | | 53±12 |
| Hand | 8±5 | ○ | 14±3 | △ | 9±1 | 5±2 | ○ | 9±3 | △ | 6±1 |
| | I$_{lg}$ (kg cm$^2$) | | | | | I$_{lg}$ (kg cm$^2$) | | | | |
| Head | 200±26 | | 186±26 | | 202±8 | 190±23 | | 199±60 | | 176±21 |
| Up-Trunk | 1632±255 | ○ | 1040±152 | △ | 1418±156 | 1563±477 | ○ | 1015±348 | | 1205±297 |
| Abdomen | 1018±237 | | 959±253 | | 1121±198 | 616±198 | | 553±143 | △ | 736±79 |
| Pelvis | 767±117 | ○ ● | 631±66 | | 606±70 | 753±207 | | 575±185 | | 739±184 |
| Thigh | 413±60 | | 403±61 | | 408±49 | 533±110 | ● | 509±118 | △ | 346±64 |
| Shank | 54±11 | ● | 56±13 | | 67±6 | 72±15 | | 97±111 | | 52±6 |
| Foot | 14±5 | ○\|● | 8±2 | △ | 1±1 | 12±4 | ● | 10±8 | △ | 37±2 |
| Arm | 32±7 | ● | 27±6 | △ | 40±3 | 30±8 | | 43±44 | | 41±13 |
| Forearm | 8±3 | ● | 10±3 | | 12±1 | 6±2 | | 10±12 | | 8±1 |
| Hand | 3±2 | ○\|● | 6±1 | | 6±1 | 2±1 | ○ | 4±2 | | 3±1 |

Significant differences ($p<0.05$) found in post-hoc analysis are reported as symbols. Where '○' indicates a significant between the proposed and the ECM method, '●' indicates a significant between the proposed and the regression modelling, and '△' indicates a significant between ECM and regression modelling.

~30–40 min per 3D scan with the amount of time decreasing to about 25 minutes as we became proficient in the protocol. Using our software, we estimated the participant-specific BSPs using the segmented scans and compared these BSP results to those found using the two comparison methods. Our approach was straightforward to implement, low cost, and produced reliable total volume estimates between repeated 3D body scans. We found that there were no significant differences between the total volume when comparing repeated scans for both male and female participants with excellent ICC values. When comparing total body mass estimates to our gold standard medical scale, we found no significant differences in mass estimates for both sexes. We found limits of agreement for males from 2.8 to -5.0kg (+1.96 SD

**Table 5. Longitudinal length estimates (mean±SD in cm) obtained using the proposed method and the ECM approach for both male and female participants.**

| Segment | Male Length (cm) | | | | Female Length (cm) | | | |
|---|---|---|---|---|---|---|---|---|
| | Proposed | | ECM | Avg. Value | Proposed | | ECM | Avg. Value |
| Head | 25.6±0.8 | ○ | 28.9±0.9 | 24.3±# | 26.0±1.3 | ○ | 28.4±1.4 | 24.6±# |
| Up-Trunk | 24.9±1.5 | ○ | 23.8±1.6 | 24.2±# | 28.3±2.9 | | 27.3±2.7 | 22.8±# |
| Abdomen | 20.2±1.9 | | 20.5±2.1 | 21.6±# | 13.6±1.0 | | 15.9±4.5 | 20.5±# |
| Pelvis | 23.4±1.2 | ○ | 18.8±1.7 | 25.2±# | 22.6±1.4 | ○ | 21.4±2.3 | *±# |
| Thigh | 46.3±1.5 | | 47.6±3.4 | 42.2±# | 47.2±2.4 | ○ | 42.6±3.6 | 36.9±# |
| Shank | 40.2±1.4 | ○ | 41.5±2.0 | 44.0±# | 38.5±2.6 | ○ | 40.4±3.0 | 43.9±# |
| Foot | 24.5±1.0 | ○ | 22.3±0.4 | 25.8±# | 23.7±1.5 | ○ | 21.6±1.3 | 22.8±# |
| Arm | 28.1±2.0 | | 28.2±1.8 | 28.2±# | 29.0±2.3 | ○ | 26.7±2.6 | 27.5±# |
| Forearm | 25.7±0.9 | ○ | 27.4±1.2 | 26.9±# | 24.8±2.5 | | 26.1±2.0 | 26.4±# |
| Hand | 15.3±1.9 | ○ | 20.0±1.3 | 18.8±# | 14.2±1.7 | ○ | 19.0±1.1 | 17.0±# |

Significant differences (p<0.05) found in post-hoc analysis are reported as symbols. Where '○' indicates a significant difference between the proposed and the ECM method. Average values of longitudinal length estimates published by (Paolo de Leva, 1996) (as shown in Table A2.11 and Fig A.3 in [4] are shown from reference. (Here # indicates no data).

to -1.96S SD) with a mean difference (bias) of -1.1kg (-1.5%). For females, we found limits of agreement of 0.64 to -6.9kg (+1.96 SD to -1.96 SD) with a mean difference (bias) of -3.1 kg (-4.4%) (p<0.001). Our proposed 3D segmentation protocol and post-processing of 3D scans worked well. Using open-source software MeshLab, we were able to segment each scan into 16 individual body segments. We found that our proposed method compared against the other two methods but there were some differences across methods for some segments and BSPs. For example, we found that the smallest body segments (e.g., foot and hand) tended to significantly differ between comparison methods across all BSPs. More so, longitudinal length and center of mass estimates were significantly different between most of the segments when comparing the 3D scanning method and ECM approach. Our work here provides the framework

**Table 6. Center of mass position from the proximal endpoints (pCOM) (mean±SD in %) estimated using the proposed method and the ECM approach for both male and female participants.**

| Segment | Male pCOM (%) | | | | Female pCOM(%) | | | |
|---|---|---|---|---|---|---|---|---|
| | Proposed | | ECM | Mean Value | Proposed | | ECM | Mean Value |
| Head | 49.6±0.9 | ○ | 46.5±1.2 | 50.0±2.2 | 48.3±1.3 | ○ | 45.4±1.8 | 48.4±# |
| Up-Trunk | 54.8±0.9 | ○ | 58.8±2.7 | 50.7±2.2 | 57.2±1.1 | | 56.6±1.9 | 50.5±# |
| Abdomen | 48.6±1.0 | ○ | 51.2±1.5 | 45.0±2.1 | 50.0±1.6 | | 48.6±9.2 | 45.1±# |
| Pelvis | 36.5±1.1 | ○ | 52.3±5.0 | 35.4±3.0 | 36.6±1.3 | ○ | 45.8±6.2 | 34.8±# |
| Thigh | 44.0±1.0 | | 44.7±3.5 | 41.0±1.9 | 44.7±1.3 | ○ | 38.2±4.8 | 46.1±# |
| Shank | 40.6±1.1 | | 40.0±1.9 | 44.0±2.8 | 40.9±0.8 | | 41.2±1.3 | 40.3±# |
| Foot | 45.1±1.9 | ○ | 39.1±2.5 | 44.2±3.7 | 44.6±2.3 | | 38.2±2.1 | 40.1±# |
| Arm | 41.9±1.4 | | 48.2±29.8 | 42.3±4.2 | 40.9±1.0 | | 41.4±3.5 | 44.0±# |
| Forearm | 41.2±1.6 | ○ | 42.9±1.6 | 42.7±3.3 | 42.7±1.5 | | 43.9±2.5 | 42.6±# |
| Hand | 40.7±1.2 | | 40.7±2.5 | 36.9±4.9 | 42.9±2.7 | | 40.6±4.1 | 35.0±# |

Significant differences (p<0.05) found in post-hoc analysis are reported as symbols. Where '○' indicates a significant difference between the proposed and the ECM method. The regression equations provided by Zatsiorsky and Seluyanov provide pCOM in mm along the longitudinal axis without providing the longitudinal length. As such we report the average values from the study adjusted to best match our definitions of pCOM (based on Table 4.4 and Table A2.5 in [4]) (Here # indicates no data).

and useful insights for the use of a Kinect V2 for 3D scanning and estimating participant-specific BSPs.

This study has several limitations. One limitation is the manual nature of segmenting the 3D scans and the reliance here on the placed landmarks. The landmarks had a high contrast between the participant and their skin, providing us with segmentation boundaries. Poor visibility of these landmarks and the tape used to define the boundaries create an opportunity for segmentation boundary errors. Although we generally found coefficients of variations and ICC values within an acceptable range between repeated segmentations, when the visibility of the segmentation boundary was poor, we had to rely on our best judgment to make the segmentation. Developing a more automatic segmentation process could reduce the time requirements in post-processing and may remove the need for the lengthy process of physically placing skin markers (15–20 min) and then using the placed markers for segmentation [55]. A second limitation is the scan duration. Each 3D scan took ~30 seconds where the participant was required to stand still. This is enough time for body sway and the lung's movement during breathing to perturb the measured volume. To minimize this effect, we asked participants to remain still and refrain from deeper breathing, but this requirement can be problematic when working with populations that may have difficulty in standing still (e.g., children, amputees, or pregnant women). Although our proposed method did work, integrating multiple cameras could reduce scan time requirements to seconds and may further improve our scanning protocol and results especially for the distal and smaller body segments that are harder to maintain still [19,44,48]. A third limitation is the lack of a gold standard criterion for comparing BSPs. Indeed, for total body mass estimates, we use the medical scale values, and these are a strong criterion for comparison. However, when we compare the BSP estimates from our proposed work to those estimated using the elliptical cylinder method (ECM) and the regression-based modelling, both of these comparison methods have their own underlying assumptions. For example, the use of 2D images in the elliptical cylinder method compared to our 3D approach to estimate the longitudinal length and proximal center of mass estimates maybe have contributed to some of the observed differences. Comparing the BSP estimates obtained using our methods to estimates obtained using medical-based scanners, such as dual-energy x-ray absorptiometry (DEXA), would provide a stronger point of comparison. This was unfortunately out of reach for us but should be considered in future evaluations of our methods.

The absence of heterogeneity in our participant pool, the smaller sample size, and the lack of a gold standard criterion for many of the BSP estimates limit the generalization of our results. Our findings suggest that the proposed protocol, software, and hardware are reliable for estimating a range of body segment parameters on humans. We find that our approach is efficient (quick and inexpensive) and the BSPs estimates are within a comparable range to our comparison methods and to literature. Our approach also provides the framework of 3D scanning for BSP estimation on humans and may bring value to those interested in this type of work and to the community at large. However, the sensitivity to the effects of differences in participant BMI, age, height, and other anthropometric characteristics on estimated BSP values is not yet clear. Although we do not have reason to believe that our methods would not work well with a more heterogeneous participant pool, further work with more participants will be required to appropriately quantity this before such conclusions can be made. Comparing the BSP estimates on our participants obtained using our methods to a gold standard criterion such as those obtained from a DEXA method would further increase the generalization.

The depth camera has certain limitations that need to also be considered. Firstly, when we encircled the Kinect V2 around our participants, their arms came significantly closer to the camera than any other body part. This may have contributed to some of the lack of texture and finer detail observed in some of the segments (Fig 8). The optimal accuracy of the Kinect V2

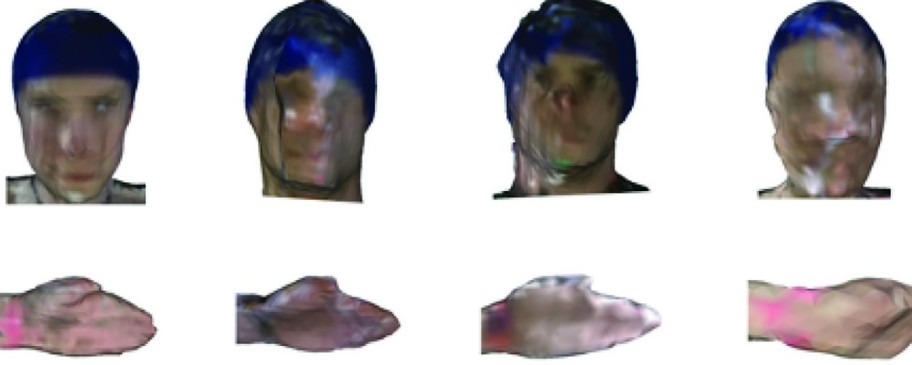

**Fig 8. Top view of full-body, head and hand scans. 1.** An example showing lack of texture in the scan on the upper arm regions. **2.** Varied degree of detail in the head region with concave surfaces appears as filled in. Further to the right head suggests minor sway, as facial features are distorted more so than the head to the left. Hands showing varied degree of detail.

depth perception is in the 0.5-2m range away from the participant, where the camera has depth perception errors of less than 2mm [56]. Scanning distances of less than 0.5m away, which would be the case for participants who had an arm's length of more than 0.5m, enter a suboptimal and more error-prone field of view [56]. 3D scanning of curved, concave, and sharp edges is also a known limitation with 3D scanning technologies. Error-prone surfaces may be smoothed out when perceiving depth [34,56]. To minimize this in our protocol, we asked participants to keep their hands open during scanning as a means of preventing concavity when the hands are supinated. However, visual occlusion of other concave surfaces is still evident, such as on the face (Fig 8). For females with larger breasts, the gap between the breasts may have been overestimated, especially as all female participants wore a shirt. As our approach estimated upper trunk masses about ~2kg higher for females (Table 5) when compared to the ECM approach, these overestimations may be partially due to sensor limitations for convex regions and gaps. Lastly, the depth perception of the device has been shown to be a function of operating time where after ~40 minutes of operating and warming up, the depth

perception reaches a steady-state value [45]. Although depth differences between using the device right away and using the device after it has been warmed up are suggested to be small (~0.003m$^2$), the effects of this were not included in our study but may be important to consider when using this device for other work.

Our work progresses on currently available tools, but further improvements need be considered before implementation. Our 3D scanning approach has the advantage that we directly measure the 3D shape, unlike regression modelling, other commonly used methods, or the elliptical cylinder method which uses 2D images to infer 3D shapes [4,25,30,57]. By working directly in 3D our approach has the added advantage that assumptions about geometry can be minimized. Although we choose to use uniform density values, working in 3D also has the added advantage that non-uniform density functions can be implemented into the workflow, speaking to the flexibility of a surface scanning approach with programmatic and easily modifiable inputs [58]. Indeed, we also found that our 3D scanning system had a measurement bias in underestimating total body mass for males (bias = -1.1 kg (-1.5%)) and females (bias = -3.1 kg (-4.4%)) when compared to the medical scale. A similar order of magnitude bias was reported for the use of older Kinect models for estimating volume and length [35,41]. In one study, the trunk BSPs estimated using a geometrical model and several other approaches were compared to gold standard DEXA. There the authors found mean differences between the gold standard and the other approaches for trunk mass, center of mass, and moments of inertia ranging from overestimating by 18.3±15.1% to underestimating by -30.2±7.1% [58]. Although the existence of a bias in our approach cannot be discounted and should be carefully considered, the range of this measurement error appears to be less than the possible errors from other widely used methods. By making our work open source we strive to provide the community with these tools and facilitate its use for further development to reduce bias and improve accuracy.

In our work here we evaluated a low-cost, easy-to-implement 3D scanning method that can be widely adapted for implementation. 3D scanning provides an exciting opportunity for estimating a wide range of body segment parameters without the need for geometric models or making predefined assumptions about the body, the age, or ethnic background of a person. As BSPs are difficult to estimate directly, especially for inner body segments like the trunk, our approach here provides a convenient solution. As we have shown, our system is able to provide users with estimates of participant-specific BSPs. The body segmentation method we developed can also be adapted to be used with 3D scans acquired using different cameras and different approaches, not limiting researchers to the use of a specific camera for 3D scanning. Although there is no limitation on how the 3D scan is acquired as any STL file can be used with our methods, future work could focus on the evaluation of other devices for acquiring 3D scans. A 3D scanning approach enables measurements and estimated BSPs to be specific to the person of interest without reliance on prior assumptions on the geometry of their bodies.

## Supporting information

**S1 File. Anatomical land marking guidelines.**
(DOCX)

**S2 File. Density values and regression equations.**
(DOCX)

**S3 File. Longitudinal length and center of mass definitions.**
(DOCX)

**S4 File. Elliptical cylinder method setup.**
(DOCX)

**S5 File. Scanning of cylindrical tube.**
(DOCX)

**S6 File. Participant volumes for repeated scans and ICC value for segmentations.**
(DOCX)

## Acknowledgments

We thank all of our participants who volunteered their time for our study and Professor
Michael Rainbow for suggestions and guidance.

## Author Contributions

**Conceptualization:** Pawel Kudzia, Genevieve Dumas.

**Data curation:** Pawel Kudzia.

**Formal analysis:** Pawel Kudzia.

**Funding acquisition:** Genevieve Dumas.

**Investigation:** Pawel Kudzia.

**Methodology:** Pawel Kudzia, Erika Jackson.

**Project administration:** Pawel Kudzia.

**Resources:** Pawel Kudzia, Erika Jackson.

**Software:** Pawel Kudzia.

**Supervision:** Genevieve Dumas.

**Validation:** Pawel Kudzia.

**Writing – original draft:** Pawel Kudzia.

**Writing – review & editing:** Pawel Kudzia, Erika Jackson, Genevieve Dumas.

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
