## [Decision Letter · Decision Letter 0]

6 Aug 2021

PONE-D-21-21310

Estimating body segment parameters from three-dimensional human body scans

PLOS ONE

Dear Dr. Kudzia,

Thank you for submitting your manuscript to PLOS ONE. After careful consideration, we feel that it has merit but does not fully meet PLOS ONE’s publication criteria as it currently stands. Therefore, we invite you to submit a revised version of the manuscript that addresses the points raised during the review process.

The reviewers found some merit in your manuscript but all three indicated that major changes are necessary before the paper can be considered for publication. The reviewers thought that the organization of the manuscript could be improved with greater clarity in the methods section. In addition, one reviewer indicated that more emphasis should to be placed on some of the prior research using this technique in order to better support the current study.

We look forward to receiving your revised manuscript.

Kind regards,

Jeremy P Loenneke

Academic Editor

PLOS ONE

Journal Requirements:

Reviewers' comments:

Reviewer's Responses to Questions

**Comments to the Author**

1. Is the manuscript technically sound, and do the data support the conclusions?

Reviewer #1: Partly

Reviewer #2: Yes

Reviewer #3: Partly

2. Has the statistical analysis been performed appropriately and rigorously? 

Reviewer #1: No

Reviewer #2: Yes

Reviewer #3: Yes

3. Have the authors made all data underlying the findings in their manuscript fully available?

Reviewer #1: Yes

Reviewer #2: Yes

Reviewer #3: Yes

4. Is the manuscript presented in an intelligible fashion and written in standard English?

Reviewer #1: Yes

Reviewer #2: Yes

Reviewer #3: Yes

5. Review Comments to the Author

Reviewer #1: The authors sought to develop a technique to accurately measure body segments using 3D human body scans. The manuscript is well written, but several things could be done to improve it.

Introduction

The authors did a nice job of showing the need for more feasible and inexpensive ways to estimate body segment parameters.

Page 69. The authors state that small variations in BSP inputs can cause clinically significant outcome measures. This is relevant to understand the acceptable amount of error that would be needed to justify using your proposed technique. If the BSP is off by 2% from the true BSP using your method, is this acceptable to not cause clinically significant differences or does the difference need to be smaller? Elaborating more on this in the discussion may benefit the readers understanding of the value of the proposed method.

Methods

How did the authors determine the sample size needed for the study?

Line 221. Are the MATLAB scripts available in a supplemental file to researchers seeking to use this method? If the authors would like this to be used in the future, this would be a valuable resource for other people seeking to use this technology.

The figures are very helpful in understanding the analysis.

Line 227. Why did the authors decide to use density inputs from older population cadavers when their population was much younger?

Line 257 and 272. One challenge with this study is the use of the elliptical cylinder method and regression model. Which one of these does the author consider as the criterion? Based on the authors introduction, it doesn’t appear that either would be a true criterion. In interpreting the results in the discussion, the authors should discuss this limitation of the study.

Lines 290-292. How were these values calculated? What reliability statistic was used?

Lines 304-305. How were the coefficient of variations calculated and what is the reference they used to label the different %s as not acceptable, acceptable, good or very good?

Line 308-309. Why didn’t the authors use other validity statistics such as mean absolute percent error or Bland-Altman analysis (limits of agreement, mean bias) to determine the validity of their new procedure to determine body mass? These other validity statistics should be reported.

Line 314-317 Why did the authors choose to not calculate the ICC values for these measurements?

Lines 330-331. The authors should have other measures of validity for their measurement besides only looking at one-way ANOVAs. Do they consider the ECM approach the “criterion” in this study?

Results

Line 409-410. The authors need to be clearer here. From the table, there are multiple body segment measurements which differ from each other. In the written part of the results, it seems as though there is no difference when the authors state that the new proposed method provided “estimates that were comparable to those determined using the ECM and regression modeling approaches”.

Discussion

Line 486 to 487. The authors need to clarify what they mean here by the body segments being comparable to the other methods. The authors state that the smallest segments had the largest differences. What does this really mean? What is the criterion you are comparing the proposed method to? Having a criterion method and using other validity statistics will help clarify the meaning of these differences you are seeing. Also, in the introduction, the authors stated that a technique that could accurately measure body segments would be helpful for amputees but if the largest differences were found in the smallest body segments (hand/foot), how helpful would this be for them? The authors should address these points.

Reviewer #2: The authors present an interesting investigation detailing the development of a procedure to estimate body segment parameters using a commercially available Microsoft Kinect camera. It is presented as an alternative to gold standard methods and existing potentially flawed approaches. According to the findings, there is some small variations with the proposed methods compared to these flawed approaches. The paper is generally well-written, but the format of the paper does seems to deviate from those within my own discipline particularly within the Methods and Results sections. This does not seem to detract from the paper and it is organized and reads well.

While a comparison to the gold standard isn’t necessarily a requirement, improvements with respect to time required, ease of use (i.e. specialized software), and need for assumptions (i.e. density) are not adequately presented or discussed making it difficult to interpret if these issues are overcome with the new method compared to other methods utilized. For example, how long did the proposed procedure take, inclusive of landmark identification and all of the body scans, and how does this compare to the other methods? Furthermore, if there are differences between the evaluated methods, is there a way to state that one is an improvement over another? Within the 3D scanning procedure, it is difficult to whether the lower or higher height scans or a mix of both were utilized in the statistical comparison. A more clear set of conclusions is needed.

The available literature on the use of the Kinetic cameras to evaluate body size/shape seems to be only briefly mentioned. A quick search yielded several references reporting biases compared to gold standard methods. With a similar premise needed to support the measurement of BSPs, this seems to be an important area that needs to be discussed.

A quick look at the demographic data provided in Table 1 appears to show that the sample was rather homogenous in nature that appears to include an “average” set of participants, while a comprehensive evaluation of the method would like likely require a more heterogenous sample with a broader set of anthropometric features.

Reviewer #3: A small number of specific comments are given below. However, detailed comments for all sections are not provided because the very small sample size is viewed as a critical flaw in the present research. If the same analytical methods could be applied to a much larger number of individuals (≥100), this research would have much greater value. In its present form, I don’t think that appropriate confidence can be placed in results based on such a small sample (relative to this field of research).

Additional comments

- The topic of the manuscript is relevant and interesting. The manuscript is well-written and informative. However, the sample size is far too small for a study like this. What is the rationale for such a small sample size (n=21)? This is a very simple data collection, and it should be feasible to attain a much larger sample. Related previous investigations have tested much larger samples. For example: Tian et al. 2020 (n>300) [PMID: 32978970], Tinsley et al. 2020 (n=179) [PMID: 31685968], Bourgeois et al. 2017 (n=113) [28876331], etc. Relevant articles cited by the authors, such as Zatsiorsky et al, used much larger numbers (n=100).

- There needs to be a better justification for the CV thresholds. It is very surprising to see 20-30% as acceptable, 10-20% as good, and <10% as very good. Where did these come from? This seems very liberal as even 10% would be considered very high for most relevant anthropometric measurements.

- In the Results, simply stating that there was no statistically significant difference between scans, based on a very small sample size, is not sufficient justification for concluding there is no (relevant) difference in total body volume between repeated scans.

- Another example of statistical significance alone not being sufficient justification is seen with the total body mass results. Even without a significant difference, the mean difference was 1.1 kg between the medical scale and the proposed method in males. This is a non-negligible amount in terms of practical purposes. The performance in females was worse, with a mean difference of almost 3 kg.

- Additional detailed comments are not provided due to this reviewer’s belief that the sample size precludes this research from being a valuable contribution to the literature. With that said, if the same analytical procedures could be repeated in a much larger sample, I think this research would make a valuable contribution.

6. PLOS authors have the option to publish the peer review history of their article (what does this mean?). If published, this will include your full peer review and any attached files.

Reviewer #1: No

Reviewer #2: No

Reviewer #3: No

---

## [Author Response · Author response to Decision Letter 0]

8 Nov 2021

We thank all the reviewers for their helpful insights, time, and feedback on our manuscript. We have done our best to reply to all of your comments as best as we could. 

REVIEWER #1 

Reviewer #1: The authors sought to develop a technique to accurately measure body segments using 3D human body scans. The manuscript is well written, but several things could be done to improve it.

Introduction

P.1 The authors did a nice job of showing the need for more feasible and inexpensive ways to estimate body segment parameters. 

Response: We thank the reviewer for this remark. 

P.2 Line 69. The authors state that small variations in BSP inputs can cause clinically significant outcome measures. This is relevant to understand the acceptable amount of error that would be needed to justify using your proposed technique. If the BSP is off by 2% from the true BSP using your method, is this acceptable to not cause clinically significant differences or does the difference need to be smaller? Elaborating more on this in the discussion may benefit the reader’s understanding of the value of the proposed method.

Response: We thank for reviewer for this comment and agree that more elaboration is needed. The most accurate BSP estimates should always be the goal but in reality, these parameters are tricky to estimate from people directly. This is especially true for the internal segments which require segmentation (e.g trunk, pelvis, thigh). The use of gold standard techniques such as DEXA is certainly not possible in most cases, nor is it practical. We have made changes in the introduction : 

“Participant-specific BSPs can reduce errors associated with biomechanical outcome measures. BSPs estimates are sensitive to the morphology, age, and gender of a person (6,7). And many biomechanical outcomes measures, such as kinetics of motion, require some BSP values to evaluate. Variabilities of +/-5% in BSP estimates can have potentially meaningful effects on the resultant outcomes (8–11). When comparing different methods used to estimate BSPs, there may also be differences in multiple of the BSPs estimates, further increasing uncertainty in the outcome measures (12). As some segment BSPs are difficult to estimate (e.g. trunk moments of inertia), using the best available tools to get representative should be the goal. The use of representative participant-specific estimates for BSPs is especially important in open-chain or high acceleration motion, such as running and jumping, where there are large body segment accelerations, and in airborne movements, where there are no external forces (13). Populations that have less available data for making approximations using ‘generic’ datasets, such as pregnant women (14), amputees (15), and children (6), may also meaningfully benefit from the use of participant-specific BSPs on outcome measure accuracy. “ 

We have also added to the discussion comparing the wide range of uncertainty in different methods for estimating BSPs. We have added the following : 

“Our work progresses on currently available tools, but further improvements need be considered before implementation. Our 3D scanning approach has the advantage that we directly measure the 3D shape, unlike regression modelling, other commonly used methods, or the elliptical cylinder method which uses 2D images to infer 3D shapes (4,25,30,53). By working directly in 3D our approach has the added advantage that assumptions about geometry can be minimized. Although we choose to us uniform density values, working in 3D also has the added advantage that non-uniform density functions can to be implemented into the workflow, speaking to the flexibility of a surface scanning approach with programmatic and easily modifiable inputs (54). Indeed, we also found that our 3D scanning system had a measurement bias in underestimating total body mass for males (bias = -1.1 kg (-1.5%)) and females (bias = -3.1 kg (-4.4%)) when compared to the medical scale. A similar order of magnitude bias was reported for the use of older Kinect models for estimating volume and length (35,55). In one study, the trunk BSPs estimated using a geometrical model and several other approaches where compared to gold standard dual-energy X-ray absorptiometry (DEXA). There the authors found mean differences between the gold standard and the other approaches for trunk mass, center of mass, and moments of inertia ranging from overestimating by 18.3±15.1% to underestimating by -30.2±7.1% (54). Although the existence of a bias in our approach cannot be discounted and should be carefully considered, the range of this measurement error appears to be less than the possible errors from other widely used methods. By making our work open source we strive to provide the community with these tools and facilitate its use for further development to reduce bias and improve accuracy.” 

It is true that higher accuracy is better but as can be seen in literature the variability between different methods can be quite high. Our method is also prone to its own errors but these errors seem to be within and lower than other approaches. As more needs to be done to better understand how this can be improved we also want to make it clear to our readers the limitations of our work here. We have added the following to the discussion : 

“Our work progresses on currently available tools, but further improvements need be considered before implementation. Our 3D scanning approach has the advantage that we directly measure the 3D shape, unlike regression modelling, other commonly used methods, or the elliptical cylinder method which uses 2D images to infer 3D shapes (4,25,30,53). By working directly in 3D our approach has the added advantage that assumptions about geometry can be minimized.”

We also include in the discussion all the limitations of our work and of the camera itself to provide the readers with the full picture. 

Methods

P.3 How did the authors determine the sample size needed for the study?

Response: The sample size of our study was based on our available resources at the time. Indeed, the sample size of other studies in literature in some cases is much larger than that of our study. A major focus and allocation of our resources were to develop and engineer the methods that are summarized in our presented paper. We believe that this work has important contributions to those interested in this field and to the readers of PlosOne. In addition, we are contributing our code and engineering work to the community at large, which many of the research papers found in the literature for work in a similar area have not been made available. We are proud of this and feel that although our study has a limited sample size ]of 21 participants we think our works still adds value. 

With a sample size of 21 participants (and 3 scans analyzed per person) in our study, we feel that the work we have done is rigorous. Nevertheless, the reviewer is right in pointing/ inferring our limited sample size. One major limitation for us is that we had a small budget and limited research resources/personal. Our resources enabled us the time to collect and process 21 participants for evaluation of our methods. As mentioned in our paper, this involved extensive post-processing per participant (30-40 minutes per scan) after we had already fine-tuned the methods. Unlike other papers where the researchers may have simply 3d scanned each participant, our research also involved placing body landmarks on each participant, and extensive post-processing to evaluate the outcomes measures/protocol. When developing this work it in fact took the authors much longer in duration per scan for initial post-processing (before we had cleaned up the methods and had figured out the most efficient process in pilot testing). 

Although a larger sample size would allow us to further evaluate our methods it is not realistic for us anymore to return to this and collect more participants. In light of the Covvid-19 pandemic and the resources available to us, we have to work with what we have. This being said, we would like the reviewer to also consider other papers with sample sizes closer to or less than our work here that have made important contributions to our field. Some examples of these publications that are related to the topic of our work include but are not limited to :

Davidson et al. 2008 (https://doi.org/10.1016/j.jbiomech.2008.09.021) had 1 participant (cited 25 times)

Sheets et al. 2010 ( https://doi.org/10.1115/1.4000155) had 4 participants (cited 11 times)

Stancic et a. 2013 (https://doi.org/10.1016/j.measurement.2012.09.010) had 8 participants (cited 29 times)

Rossi et al. 2013 (PMID: 24421737) had 28 participants (cited 29 times)

Peyer et al. 2015 (PM 25780778 ) had 6 participants (cited 19 times)

Chiu et al. 2016 (DOI: 10.1080/00140139.2016.1161245) had 17 participants (cited 6 times)

Chiu et al. 2016 (https://doi.org/10.1049/iet-smt.2015.0252) had 16 participants (cited 3 times)

Robert et al.2017 (https://doi.org/10.1080/10255842.2017.1382920)had 9 participants (cited 3 times). 

Chiu et al. 2021 (https://doi.org/10.1080/17461391.2021.1921041) had 18 participants (no citations), 

We hope that the reviewer can consider all of this when carefully considering our research. 

To increase transparency to our readers we have added the following statement in the discussion to address the sample size limitations clearly in our manuscript : 

“The lack of heterogeneity in our participant pool and the small sample size may limit the generalization of our results. Our findings suggest that the BSP estimates we obtain using the proposed protocol, software, and hardware are reliable for estimating a range of body segment parameters on humans. Our methods provide a framework and may bring value to those interested in this type of work and to the community at large. However, the effects of differences in participant BMI, age, height, and other anthropometric characteristics on estimated BSP values are not yet clear. Although we do not have reason to believe that our methods would not work well with a more heterogeneous participant pool, further work with more participants will be required to appropriately quantity this.”

P.4 Line 221. Are the MATLAB scripts available in a supplemental file to researchers seeking to use this method? If the authors would like this to be used in the future, this would be a valuable resource for other people seeking to use this technology.

Response: We agree with the reviewer. We have made all the files needed in evaluating body segment parameters from 3d body scans publicly available through Github. The software uses any STL file as an input. We hope that this can be a useful resource the community can benefit from and evolve. 

P.5 The figures are very helpful in understanding the analysis.

Response: We thank the reviewer for this kind remark. 

P.6 Line 227. Why did the authors decide to use density inputs from older population cadavers when their population was much younger?

Response: The density inputs that we chose to use for this study reflect commonly used values from the literature. Ideally, we would want to use density inputs that best reflect our sample pool but there is a scarcity of this data available. One benefit from our methods is that the density values are easily modifiable inputs that those seeking to use our methods can change to best meet their needs. It is not uncommon to use a general density value across all of the segments of 1000kg/m3. As such to see if the estimated total body mass would be better estimated using the value for the density of water for all the segments. We have added the following to the methods:

“To minimize possible effects and assumptions associated with our use different estimated density values for each body segment to calculate total body mass, we also evaluated the limits of agreement and bias using a standard uniform density value of 1000 kg/ m3 across all body segments (density of water).”

And the following to the results:

“Performing this analysis using a constant density of 1000 kg/m3 across all body segments resulted in results that led to the same conclusions. “ 

P.7 Line 257 and 272. One challenge with this study is the use of the elliptical cylinder method and regression model. Which one of these does the author consider as the criterion? Based on the author’s introduction, it doesn’t appear that either would be a true criterion. In interpreting the results in the discussion, the authors should discuss this limitation of the study.

Response: We thank the reviewer for this remark. The reviewer makes a good point regarding criterion. The elliptical cylinder method is prone to errors of its own and the regression model, as mentioned in the introduction, also has its own. We compared the data that we have collected and acknowledged that there is no true criterion to compare too. Indeed, such a criterion would be difficult to acquire given the nature of body segment parameter estimation. The only true criterion we have is the total body mass estimates from the medical scale. To make this clear to our readers we have added the following in the discussion and addressed this limitation clearly:

“...A third limitation is the lack of a gold standard criterion for comparing BSPs. Indeed, for total body mass estimates, we use the medical scale values, and these are a strong criterion for comparison. However, when we compare the BSP estimates from our proposed work to those estimated using the elliptical cylinder method and the regression-based modelling, both of these comparison methods have their own underlying assumptions. For example, the use of 2D images in the elliptical cylinder method compared to our 3D approach to estimate the longitudinal length and proximal center of mass estimates maybe have contributed to some of the observed differences. Comparing the BSP estimates obtained using our methods to estimates obtained using medical-based scanners, such as dual-energy x-ray absorptiometry (DEXA), would provide a stronger point of comparison. This was unfortunately out of reach for us but should be considered in future evaluations of our methods.” 

P.8 Lines 290-292. How were these values calculated? What reliability statistic was used?

Response: To calculate the values that determined if we overestimated or understated, we compared the predicted values to the mathematical expression of the beam (geometrical expression). Using the mathematically calculated values for volume, pCOM, etc. we then determined how much our method predictions differ from the expression. We include equations and data in supplementary file S5 for readers and modified the text to improve clarity within this paragraph. 

“To get an approximation of the relative accuracy of the device and verify that our approach was working as intended we scanned a cylindrical beam 25 times using a modified version of our 3D scanning protocol outlined above (for beam values and calculations see S5 File). We determined the mathematical geometrical expressions for the inertial parameters of the beam including total volume, longitudinal length, the proximal center of mass position, and the mass moments of inertia in the three orthogonal principal axes of the beam. We found that the 3D scanning estimates were within a reasonable range when compared to the theoretical predictions. For example, when comparing the total volume of the beam using the mathematical expression (4964 cm3) to our methods (5173±204cm3) our 3D scanning method overestimated volume on average by +4.2%. When comparing the longitudinal length (expression: 94.1cm; our methods 93.6±20.7cm) and pCOM (expression: 50%; our methods: 49.8±0.9%) our approach on average underestimated length by approximately -0.6% and pCOM by -0.3%. The orthogonal mass moments of inertia approximated using our approach differed on average by less than +1% (anteroposterior axes Iap = expression: 3676 kg/cm2; our methods 3707±205 kg/cm2 ; mediolateral axes Iml =expression: 3676 kg/cm2; our methods 3702±205 kg/cm2) with the largest difference observed for the longitudinal axes of the beam of +7.8% (longitudinal axes Ilong = expression: 42 kg/cm2; our methods 45.3±4.6 kg/cm2). This experiment gave us confidence that the outputs we found were within a reasonable range of what we expected to find.”

P.9 Lines 304-305. How were the coefficients of variations calculated and what is the reference they used to label the different %s as not acceptable, acceptable, good or very good?

Response: We thank the reviewer for this question and for the important clarification that needs to be addressed. The coefficient of variation was calculated by taking the mean of the parameter and dividing it by the standard deviation then multiplying this by 100 to express it in a unitless measure of %. We have added this equation to the body of the text. 

We agree with the reviewer that our chosen cut-off values for CV are somewhat arbitrary. There does not appear to be a clear consensus on accepted values for CV in the sports sciences and engineering fields ( for example DOI 0112-1642/98/0010-0217/$11.00/0 , Shechtman, O. (2013). The Coefficient of Variation as an Index of Measurement Reliability ). We agree that our previously chosen cut-off values are too generous and based on the literature we have changed this in the text and replaced it with the following values: 

 “Following recommended guidelines we considered ICC (2,1) = < 0.5 as poor, 0.50-0.75 as moderate, 0.75-0.9 as good, and >0.9 as excellent (46). We calculated the coefficients of variations (CV = mean/standard deviation x 100) of body volume estimation to express a measure of normalized variability between repeated scans. We considered coefficients of variations >15% as not acceptable, 15-10% as acceptable, 10-5% as good, and <5% as very good. As we could not find a consensus for acceptable values for coefficients of variation and arbitrarily determined acceptable values widely range between fields of research (47). We, therefore, based our considerations using a commonly reported cut-off value of 15%.” 

P.10 Line 308-309. Why didn’t the authors use other validity statistics such as mean absolute percent error or Bland-Altman analysis (limits of agreement, mean bias) to determine the validity of their new procedure to determine body mass? These other validity statistics should be reported.

Response: We thank the reviewer for bringing this up and agree that other validity statistics would be beneficial. We have now reported the Bland-Altman analysis. We have added the following to the methods:

“We also evaluated the total body mass agreements between our 3D scanning estimates and the medical scale using a Bland-Altman approach (48). Here we found the limits of agreement by comparing the differences between the two methods and report these limits along with any found bias. A positive bias is an indication that the 3D scanning approach overestimates mass whereas a negative bias is an indication of underestimating mass” 

In the results we have added the following:

“When comparing the total body mass predictions from all of our 3D scans to the medical scale mass for males, we found limits of agreement from 2.8 to -5.0kg (+1.96 SD to -1.96S SD) with a mean difference (bias) of -1.1kg. For females, we found limits of agreement of 0.64 to -6.9kg (+1.96 SD to -1.96 SD) with a mean difference (bias) of -3.1 kg (p<0.001)” 

P.11 Line 314-317 Why did the authors choose to not calculate the ICC values for these measurements?

Response: We thank the reviewers for pointing this out. We have now calculated and provided the results of the ICC values across the segmentations for all of the body segments and estimated BSPs. We have added the following to the methods:

“Following the same approach as with the total body volume, here we also calculated the 2-way mixed-effects intraclass correlation coefficients (ICC) to provide estimates in the consistency of the estimated BSPs across segmentations” 

And have added the following to the results: 

“In most instances, we found high ICC estimates for all of the evaluated BSPs across repeated segmentations corresponding to excellent reliability (ICC (2,1)>0.9) (see S6 File for full table). We did however find that for a few BSPs and for certain body segments the ICC estimates were poor (ICC(2,1)<0.5), suggesting poor repeatability. This was observed for both males and females and in most cases for the smaller distal body segments specifically the foot, hand, and arm” 

And include the table of all ICC values in the Appendix 

P.12 Lines 330-331. The authors should have other measures of validity for their measurement besides only looking at one-way ANOVAs. Do they consider the ECM approach the “criterion” in this study?

Response: We have added the limitation regarding criterion as stated above in:

“.. A third limitation..” and in the preceding paragraph in the discussion:”...The absence of heterogeneity in our participant pool, the smaller sample size, and the lack of a gold standard criterion for many of the BSP estimates limit the generalization of our results..”

As we state in this limitations section of the discussion we do not have a strong gold standard for the BSP measurements to compare to. ECM is a good measure but it is a 2D method and regression modelling is also just its own approximation. BSPs in general can be difficult to compare as acquiring a gold standard estimate (such as those obtained using medical imaging) is difficult, especially when resources are limited as was in our case. We, therefore, feel that the summary tables we provide (Table 3-6) for our BSP estimates across the three methods and the accompanying results of the repeated measures analysis provide the readers with a strong sense of where our method does well and where it still needs improvement. If the reviewer has a strong suggestion for further statistical analysis that would add to the readers of our paper, we very much are open to the suggestion and appreciate recommendations. 

Results

P.13 Line 409-410. The authors need to be clearer here. From the table, there are multiple body segment measurements that differ from each other. In the written part of the results, it seems as though there is no difference when the authors state that the newly proposed method provided “estimates that were comparable to those determined using the ECM and regression modelling approaches”.

Response: We thank the reviewer for the feedback and agree that the clarity here can be improved. To better present the results from our study we have split up this section into two sections in the results. The first section is now the mass and mass moments of inertia which had fewer differences across methods and body segments than the second section which is the lengths and centres of masses estimates. By splitting it up into two sections we hope we now better address the lack of clarity the reviewer has pointed out. We have also made small changes in the text to improve the clarity. 

Discussion

P14. Line 486 to 487. The authors need to clarify what they mean here by the body segments being comparable to the other methods. The authors state that the smallest segments had the largest differences. What does this really mean? What is the criterion you are comparing the proposed method to? Having a criterion method and using other validity statistics will help clarify the meaning of these differences you are seeing. 

Response: The reviewer brings up an important point and we agree that further clarification is needed in our manuscript. To address these points we have changed the first paragraph in the discussion to read as follows: “

“We evaluated an inexpensive 3D surface scanning approach for estimating participant-specific BSPs. We used a readily available consumer depth camera, the Kinect V2 to collect repeated 3D body scans of 21 participants. Interaction with the participant for acquiring the 3D scan took around 20 minutes (broken down to between 15-20 minutes for landmarking, and 30 seconds per scan). The post-processing from importing the 3D scan to outputted BSPs took ~30-40 min per 3D scan with the amount of time decreasing to about 25 minutes as we became proficient in the protocol. Using our software, we estimated the participant-specific BSPs using the segmented scans and compared these BSP results to those found using the two comparison methods. Our approach was straightforward to implement, low cost and produced reliable total volume estimates between repeated 3D body scans. We found that there were no significant differences between the total volume when comparing repeated scans for both male and female participants with excellent ICC values. When comparing total body mass estimates to our gold standard medical scale, we found no significant differences in mass estimates for both sexes. We found limits of agreement for males from 2.8 to -5.0kg (+1.96 SD to -1.96S SD) with a mean difference (bias) of -1.1kg (-1.5%). For females, we found limits of agreement of 0.64 to -6.9kg (+1.96 SD to -1.96 SD) with a mean difference (bias) of -3.1 kg (-4.4%) (p<0.001). Our proposed 3D segmentation protocol and post-processing of 3D scans worked well. Using open-source software MeshLab, we were able to segment each scan into 16 individual body segments. We found that our proposed method compared against the other two methods but there were some differences across methods for some segments and BSPs. For example, we found that the smallest body segments (e.g., foot and hand) tended to significantly differ between comparison methods across all BSPs. More so, longitudinal length and center of mass estimates were significantly different between most of the segments when comparing the 3D scanning method and ECM approach. Our work here provides the framework and useful insights for the use of a Kinect V2 for 3D scanning and estimating participant-specific BSPs.”

We have also added and clarified the limitations of our work more clearly: ”

“The absence of heterogeneity in our participant pool, the smaller sample size, and the lack of a gold standard criterion for many of the BSP estimate limit the generalization of our results. Our findings suggest that the proposed protocol, software, and hardware are reliable for estimating a range of body segment parameters in humans. We find that our approach is efficient and the BSPs estimates are within a comparable range to our comparison methods and to literature. Our approach also provides the framework of 3D scanning for BSP estimation in humans and may bring value to those interested in this type of work and to the community at large. However, the effects of differences in participant BMI, age, height, and other anthropometric characteristics on estimated BSP values are not yet clear. Although we do not have reason to believe that our methods would not work well with a more heterogeneous participant pool, further work with more participants will be required to appropriately quantity this before any conclusions can be made. Comparing the BSP estimates on our participants obtained using our methods to a gold standard criterion such as those obtained from a DEXA method would further increase the generalization. “

P15. Also, in the introduction, the authors stated that a technique that could accurately measure body segments would be helpful for amputees but if the largest differences were found in the smallest body segments (hand/foot), how helpful would this be for them? The authors should address these points.

Response: We believe one source of error for the distal segment is the duration of the scan. At its current state, it is likely that our system is not yet accurate enough for evaluation on these distal segments. We address this in the discussion throughout but also in the following :

“A second limitation is the scan duration. Each 3D scan took ~30 seconds where the participant was required to stand still. This is enough time for body sway and the lung’s movement during breathing to perturb the measured volume. To minimize this effect, we asked participants to remain still and refrain from deeper breathing, but this requirement can be problematic when working with populations that may have difficulty in standing still (e.g., children, amputees, or pregnant women). Although our proposed method did work, integrating multiple cameras could reduce scan time requirements to seconds and may further improve our scanning protocol and results especially for the distal and smaller body segments (19,42,51).”

REVIEWER #2

Reviewer #2: The authors present an interesting investigation detailing the development of a procedure to estimate body segment parameters using a commercially available Microsoft Kinect camera. It is presented as an alternative to gold-standard methods and existing potentially flawed approaches. According to the findings, there are some small variations with the proposed methods compared to these flawed approaches. The paper is generally well-written, but the format of the paper does seem to deviate from those within my own discipline particularly within the Methods and Results sections. This does not seem to detract from the paper and it is organized and reads well.

Response: We thank the reviewer for taking the time and reviewing our paper. 

P.1 While a comparison to the gold standard isn’t necessarily a requirement, improvements with respect to time required, ease of use (i.e. specialized software), and need for assumptions (i.e. density) are not adequately presented or discussed making it difficult to interpret if these issues are overcome with the new method compared to other methods utilized. For example, how long did the proposed procedure take, inclusive of landmark identification and all of the body scans, and how does this compare to the other methods?

Response: We thank the reviewer for this comment and agree that further clarification in many of these points needs to be added to our manuscript to further improve it for our readers. The placing of landmarks took around 15-20 minutes, the scan duration was about 30 seconds, and the post-processing took around 30-40 min with this time decreasing as our proficiency in the protocol developed. All in all, this means that from scanning to output it took ~ 1 hour with time with the participant being about 20 minutes. To ensure full transparency to the reader we include these times in our discussion and have now edited and added the following: 

“Interaction with the participant for acquiring the 3D scan took around 20 minutes (broken down to between 15-20 minutes for landmarking, and 30 seconds per scan). The post-processing from importing the 3D scan to outputted BSPs took ~30-40 min per 3D scan with the amount of time decreasing to about 25 minutes as we became proficient in the protocol.” 

“A second limitation is the scan duration. Each 3D scan took ~30 seconds where the participant was required to stand still. This is enough time for body sway and the lung’s movement during breathing to perturb the measured volume. To minimize this effect, we asked participants to remain still and refrain from deeper breathing, but this requirement can be problematic when working with populations that may have difficulty in standing still (e.g., children, amputees, or pregnant women). Although our proposed method did work, integrating multiple cameras could reduce scan time requirements to seconds and may further improve our scanning protocol and results especially for the distal and smaller body segments (19,42,51)”

“Our work progresses on currently available tools, but further improvements need be considered before implementation. Our 3D scanning approach has the advantage that we directly measure the 3D shape, unlike regression modelling, other commonly used methods, or the elliptical cylinder method which uses 2D images to infer 3D shapes (4,25,30,53). By working directly in 3D our approach has the added advantage that assumptions about geometry can be minimized. Although we choose to use uniform density values, working in 3D also has the added advantage that non-uniform density functions can be implemented into the workflow, speaking to the flexibility of a surface scanning approach with programmatic and easily modifiable inputs (54). Indeed, we also found that our 3D scanning system had a measurement bias in underestimating total body mass for males (bias = -1.1 kg (-1.5%)) and females (bias = -3.1 kg (-4.4%)) when compared to the medical scale. A similar order of magnitude bias was reported for the use of older Kinect models for estimating volume and length (35,55). In one study, the trunk BSPs estimated using a geometrical model and several other approaches were compared to gold standard DEXA. There the authors found mean differences between the gold standard and the other approaches for trunk mass, center of mass, and moments of inertia ranging from overestimating by 18.3±15.1% to underestimating by -30.2±7.1% (54). Although the existence of a bias in our approach cannot be discounted and should be carefully considered, the range of this measurement error appears to be less than the possible errors from other widely used methods. By making our work open source we strive to provide the community with these tools and facilitate its use for further development to reduce bias and improve accuracy.”

To give the reviewer an idea of the time duration of other methods available in the literature we looked at a lot of papers and in many cases, the authors provide vague time estimates such as (low time) or do not provide the full breakdown. Here are a few highlights from our findings : 

[S. H. L. Smith and A. M. J. Bull, “Rapid calculation of bespoke body segment parameters using 3D infra-red scanning,” Med. Eng. Phys., vol. 62, pp. 36–45, Dec. 2018. 

60s for a scan using the method outlined in this study, followed by 10–15 min for manually finding landmarks on the point cloud, and less than minute post-processing for the BSP calculation. 

S. Clarkson, S. Choppin, J. Hart, B. Heller, and J. Wheat, “Calculating body segment inertia parameters from a single rapid scan using the microsoft kinect,” in Proceedings of the 3rd international conference on 3D body scanning technologies, 2012, pp. 153–163.

2. A single scan of the whole body takes around 3 seconds, plus the time taken to initially palpate the body. In contrast, the manual measurements required of Yeadon’s geometric model can take around 40 minutes of the subject’s time. 

J. C. K. Wells, A. Ruto, and P. Treleaven, “Whole-body three-dimensional photonic scanning: a new technique for obesity research and clinical practice,” Int. J. Obes. , vol. 32, no. 2, pp. 232–238, Feb. 2008. 

3. They just state: Low time 

K. E. Peyer, M. Morris, and W. I. Sellers, “Subject-specific body segment parameter estimation using 3D photogrammetry with multiple cameras,” PeerJ, vol. 3, p. e831, Mar. 2015

4. The procedure is currently moderately time-consuming in total (post-processing)

5. interaction time with the participant is extremely short and involves no contact, which can be very beneficial for certain experimental protocols or with specific vulnerable participants”

In our manuscript, we provide full transparency and a breakdown of how much our methods take. Compared to other studies we believe we compare well. We also provide recommendations for improving our speed (reducing scanning time and automating the post-processing) and mention this in our limitations and future work. 

P2. Furthermore, if there are differences between the evaluated methods, is there a way to state that one is an improvement over another? 

Response: The main advantage of the methods we propose and evaluate is that it is in 3D. There are no assumptions made regarding geometry (as in 2D methods) or any modelling required to estimate BSPs (as in regression and some geometrical approaches). We have made this more throughout the discussion section of our revised manuscript. 

For example “..However, when we compare the BSP estimates from our proposed work to those estimated using the elliptical cylinder method (ECM) and the regression-based modelling, both of these comparison methods have their own underlying assumptions. For example, the use of 2D images in the elliptical cylinder method compared to our 3D approach to estimate the longitudinal length and proximal center of mass estimates maybe have contributed to some of the observed differences.”

P3. Within the 3D scanning procedure, it is difficult to whether the lower or higher height scans or a mix of both were utilized in the statistical comparison. A more clear set of conclusions is needed. 

Response: We thank the reviewer for this comment. Each scan consisted of two revolutions around the participant (but it was only one scan). One revolution where the camera was lower and one where the camera was higher by about a meter. As mentioned in our paper we choose this approach as in pilot experiments we found that to get more visually complete scans it helped to raise the camera above the participant. To increase the clarity of this to the readers we have added the following to the manuscript : 

“As a result, each complete participant scan that we used for analysis consisted of these two aforementioned revolutions.” 

P.4 The available literature on the use of the Kinect cameras to evaluate body size/shape seems to be only briefly mentioned. A quick search yielded several references reporting biases compared to gold standard methods. With a similar premise needed to support the measurement of BSPs, this seems to be an important area that needs to be discussed.

Response: Thank you for the comment and we agree that more needs to be mentioned in our introduction. It is important to note that the majority of references in the literature are regarding the Kinect V1 (some examples include: 

S. Clarkson, S. Choppin, J. Hart, B. Heller, and J. Wheat, “Calculating body segment inertia parameters from a single rapid scan using the microsoft kinect,” in Proceedings of the 3rd international conference on 3D body scanning technologies, 2012, pp. 153–163.

C.-Y. Chiu, S. Fawkner, S. Coleman, and R. Sanders, “Automatic calculation of personal body segment parameters with a microsoft kinect device”.

R. Buffa et al., “A new, effective and low-cost three-dimensional approach for the estimation of upper-limb volume,” Sensors , vol. 15, no. 6, pp. 12342–12357, May 2015.

J. Kongsro, “Estimation of pig weight using a Microsoft Kinect prototype imaging system,” Comput. Electron. Agric., vol. 109, pp. 32–35, Nov. 2014.

F. Öhberg, A. Zachrisson, and Å. Holmner-Rocklöv, “Three-Dimensional Camera System for Measuring Arm Volume in Women with Lymphedema Following Breast Cancer Treatment,” Lymphat. Res. Biol., vol. 12, no. 4, pp. 267–274, Dec. 2014.

J. Tong, J. Zhou, L. Liu, Z. Pan, and H. Yan, “Scanning 3D full human bodies using Kinects,” IEEE Trans. Vis. Comput. Graph., vol. 18, no. 4, pp. 643–650, Apr. 2012.

S. Clarkson, J. Wheat, B. Heller, and S. Choppin, “Assessment of a Microsoft Kinect-based 3D scanning system for taking body segment girth measurements: a comparison to ISAK and ISO standards,” J. Sports Sci., vol. 34, no. 11, pp. 1006–1014, 2016.

Y. Cui and D. Stricker, “3D body scanning with one Kinect,” in 2nd International Conference on 3D Body Scanning Technologies, 2011, vol. 10. [Online]. Available: http://www.3dbodyscanning.org/cap/papers/2011/11121_07cui.pdf

S. H. L. Smith and A. M. J. Bull, “Rapid calculation of bespoke body segment parameters using 3D infra-red scanning,” Med. Eng. Phys., vol. 62, pp. 36–45, Dec. 2018.

There are important differences between the sensors Kinect V1 and Kinect V2. For example, the Kinect V2 uses the time of flight technology compared to the structured light imaging for 3D data acquisition in the Kinect V1. The Kinect V2 has been evaluated to be a more accurate and overall superior sensor when compared to the V1 (please see: O. Wasenmüller and D. Stricker, “Comparison of Kinect V1 and V2 Depth Images in Terms of Accuracy and Precision,” Computer Vision – ACCV 2016 Workshops. pp. 34–45, 2017. doi: 10.1007/978-3-319-54427-4_3.). However, with the lack of case uses of the Kinect V2 it is difficult to directly comment on bias and accuracy compared to the aforementioned studies which use the Kinect V1. 

Several studies in the literature have indeed used the Kinect V2 for data collection of studies that are of a similar nature to our work (for example : 

A. J. Das, D. C. Murmann, K. Cohrn, and R. Raskar, “A method for rapid 3D scanning and replication of large paleontological specimens,” PLoS One, vol. 12, no. 7, p. e0179264, Jul. 2017.

M. Kowalski, J. Naruniec, and M. Daniluk, “Livescan3D: A Fast and Inexpensive 3D Data Acquisition System for Multiple Kinect v2 Sensors,” in 2015 International Conference on 3D Vision, Oct. 2015, pp. 318–325.

We have added the following to the introduction to address some of the literature and provide more clarity to our readers : 

“ 3D surface scanning provides an opportunity for acquiring the 3D geometry without using a geometrical model. 3D surface scanning techniques using laser scanning (32), structured light projection (33) and time of flight cameras (34) provide the tools to 3D reconstruct objects, humans, and other animals. 3D surface scanning omits the use of predefined geometrical shapes to estimate the morphology of the body and as a result does not require the use of a 2D photographic method to make anthropometric measures. The Microsoft Kinect Version 1 (Kinect V1, Microsoft Corporation, Redmond, USA) is a low-cost close-range camera that has shown potential for 3D volume estimation (35), for estimating participant-specific anthropometric measurements (36), and in some preliminary work in estimating body segment parameters (37–39). Volumetric estimations using the Kinect V1 have been reported to have errors of 0.04±2.11%, suggesting greater accuracy than commonly used geometric models (38). When comparied gold standard medical imaging to those estimated using an array of Kinect V1 cameras (16 cameras in total) a high correlation in total body volume estimation was found (R2 =0.99) but the Kinect tended to underestimate volume (40,41). Other 3D cameras have also shown promise in this field of research (19,42–44). The newest version the Kinect Version 2 (Kinect V2, Microsoft Corporation, Redmond, USA) is more accurate than the Kinect V1 in terms of depth perception and 3D estimation and boasts a higher resolution (34,45,46). In one recent study, complex dinosaur skulls were 3D scanned with the Kinect V2 and the device was found to perform as well (reported depth resolution of 0.6mm) as industrial-grade laser scanners that cost exponentially more. A consumer depth camera, such as the Microsoft Kinect V2 presents an opportunity to develop and evaluate an inexpensive approach for estimating participant-specific BSPs while addressing some of the limitations of the aforementioned methods.” 

P.5 A quick look at the demographic data provided in Table 1 appears to show that the sample was rather homogenous in nature that appears to include an “average” set of participants, while a comprehensive evaluation of the method would like likely require a more heterogeneous sample with a broader set of anthropometric features.

Response: We thank the reviewer for this comment. The reviewer is correct in stating that a more heterogeneous sample with a border range of anthropometric features is required for further evaluation of our proposed method. In its current form, we have focused on the development and testing of a novel method for estimating body segment parameters from 3D body scans and making this method available for the community. To clarify these aforementioned concerns and limitations in our work we had added the following to the manuscript in the discussion “ 

“The absence of heterogeneity in our participant pool, the smaller sample size, and the lack of a gold standard criterion for many of the BSP estimates limit the generalization of our results. Our findings suggest that the proposed protocol, software, and hardware are reliable for estimating a range of body segment parameters on humans. We find that our approach is efficient and the BSPs estimates are within a comparable range to our comparison methods and to literature. Our approach also provides the framework of 3D scanning for BSP estimation on humans and may bring value to those interested in this type of work and to the community at large. However, the effects of differences in participant BMI, age, height, and other anthropometric characteristics on estimated BSP values are not yet clear. Although we do not have reason to believe that our methods would not work well with a more heterogeneous participant pool, further work with more participants will be required to appropriately quantity this before any conclusions can be made. Comparing the BSP estimates on our participants obtained using our methods to a gold standard criterion such as those obtained from a DEXA method would further increase the generalization.”

REVIEWER #3 

Reviewer #3: A small number of specific comments are given below. However, detailed comments for all sections are not provided because the very small sample size is viewed as a critical flaw in the present research. If the same analytical methods could be applied to a much larger number of individuals (≥100), this research would have much greater value. In its present form, I don’t think that appropriate confidence can be placed in results based on such a small sample (relative to this field of research).

P.1 The topic of the manuscript is relevant and interesting. The manuscript is well-written and informative. However, the sample size is far too small for a study like this. What is the rationale for such a small sample size (n=21)? This is a very simple data collection, and it should be feasible to attain a much larger sample. Related previous investigations have tested much larger samples. For example: Tian et al. 2020 (n>300) [PMID: 32978970], Tinsley et al. 2020 (n=179) [PMID: 31685968], Bourgeois et al. 2017 (n=113) [28876331], etc. Relevant articles cited by the authors, such as Zatsiorsky et al, used much larger numbers (n=100).

Response: We thank the reviewer for this comment. Indeed, the sample size of the studies the author has mentioned is much larger than that of our study. A major focus and allocation of our resources was to develop and engineer the methods that are summarized in our presented paper. We believe that this work we present has important contributions to those interested in this field and to the readers of PlosOne. In addition, we are contributing our code and engineering work to the community at large, which many of the aforementioned methods do not do. We are proud of this and feel that although our study has some important limitations (for example smaller sample size) it still adds value. 

With a sample size of 21 participants (3 scans analyzed per person) in our study, we feel that the work we have done is rigorous. Nevertheless, the reviewer is right in pointing out that this is a limitation. One limitation for us is that we had a small budget and limited research resources. Our resources enabled us the time to collect and process 21 participants for evaluation of our methods. As mentioned in our paper, this involved extensive post-processing per participant after we had already fine-tuned the methods. Unlike other papers the reviewer had mentioned where the researcher simply 3d scanned each participant, our research also involved placing body landmarks on each participant, and extensive post-processing to evaluate our methods. When developing this work it in fact took the authors much longer than 30-40 minutes in duration per scan for initial post-processing (before we had cleaned up the methods and had figured out the most efficient process). More so, the reviewer mentions important work from others (such as Zatsiorsky et al. which we reference and compare too) where the sample size is much larger. Although a larger sample size would allow us to further evaluate our methods it is not realistic for us anymore to return to this and collect more participants. In light of the Covvid-19 pandemic and the resources available to us, we have to work with what we have. This being said, we would like the reviewer to also consider other papers with sample sizes closer to or less than our work here that have made important contributions to our field. Some examples of these publications that are related to the topic of our work include but are not limited to :

Davidson et al. 2008 (https://doi.org/10.1016/j.jbiomech.2008.09.021) had 1 participant (cited 25 times)

Sheets et al. 2010( https://doi.org/10.1115/1.4000155) had 4 participants (cited 11 times)

Stancic et a. 2013 (https://doi.org/10.1016/j.measurement.2012.09.010) had 8 participants (cited 29 times)

Rossi et al. 2013 (PMID: 24421737) had 28 participants (cited 29 times)

Peyer et al. 2015 (PM 25780778 ) had 6 participants (cited 19 times)

Chiu et al. 2016 (DOI: 10.1080/00140139.2016.1161245) had 17 participants (cited 6 times)

Chiu et al. 2016 (https://doi.org/10.1049/iet-smt.2015.0252) had 16 participants (cited 3 times)

Robert et al.2017 (https://doi.org/10.1080/10255842.2017.1382920)had 9 participants (cited 3 times). Chiu et al. 2021 (https://doi.org/10.1080/17461391.2021.1921041) had 18 participants (no citations), 

We hope that the reviewer can consider all of this when carefully considering our research. To increase transparency to our readers we have added the following statement in the discussion to address the sample size limitations clearly in our manuscript: 

“The absence of heterogeneity in our participant pool, the smaller sample size, and the lack of a gold standard criterion for many of the BSP estimates limit the generalization of our results. Our findings suggest that the proposed protocol, software, and hardware are reliable for estimating a range of body segment parameters on humans. We find that our approach is efficient (quick and inexpensive) and the BSPs estimates are within a comparable range to our comparison methods and to literature. Our approach also provides the framework of 3D scanning for BSP estimation on humans and may bring value to those interested in this type of work and to the community at large. However, the sensitivity to the effects of differences in participant BMI, age, height, and other anthropometric characteristics on estimated BSP values is not yet clear. Although we do not have reason to believe that our methods would not work well with a more heterogeneous participant pool, further work with more participants will be required to appropriately quantity this before such conclusions can be made. Comparing the BSP estimates on our participants obtained using our methods to a gold standard criterion such as those obtained from a DEXA method would further increase the generalization.”

P.2 There needs to be a better justification for the CV thresholds. It is very surprising to see 20-30% as acceptable, 10-20% as good, and <10% as very good. Where did these come from? This seems very liberal as even 10% would be considered very high for most relevant anthropometric measurements.

Response: We thank the reviewer for this important remark. After further discussion with our team and reviewing the literature we agree with the reviewer that the CV thresholds we mention and use in our paper are too liberal. 

More so, we agree with the reviewer that our chosen cut-off values for CV are somewhat arbitrary. There does not appear to be a clear consensus on accepted values for CV in the sports sciences and engineering fields (for example DOI 0112-1642/98/0010-0217/$11.00/0 , Shechtman, O. (2013). The Coefficient of Variation as an Index of Measurement Reliability ). We agree that our previously chosen cut-off values are too generous and based on the literature we have changed this in the text and replaced it with the following values : 

“Following recommended guidelines, we considered ICC (2,1) = < 0.5 as poor, 0.50-0.75 as moderate, 0.75-0.9 as good, and >0.9 as excellent (46). We calculated the coefficients of variations (CV = mean/standard deviation x 100) of body volume estimation to express a measure of normalized variability between repeated scans. We considered coefficients of variations >15% as not acceptable, 15-10% as acceptable, 10-5% as good, and <5% as very good. As we could not find a consensus for acceptable values for coefficients of variation and arbitrarily determined acceptable values widely range between fields of research (47). We, therefore, based our considerations using a commonly reported cut-off value of 15%.” 

P.3 In the Results, simply stating that there was no statistically significant difference between scans, based on very small sample size, is not sufficient justification for concluding there is no (relevant) difference in total body volume between repeated scans.

Response: We have changed the writing in the results section to better summarize our findings and improve clarity. In addition to this, we have added some additional analyses such as Blant-Altman and ICC values for repeated segmentation of volume. We hope that this improves clarity for our readers and strengthens the value of our contribution. 

P.4 Another example of statistical significance alone not being sufficient justification is seen with the total body mass results. Even without a significant difference, the mean difference was 1.1 kg between the medical scale and the proposed method in males. This is a non-negligible amount in terms of practical purposes. The performance in females was worse, with a mean difference of almost 3 kg.

Response: We thank the reviewer for pointing out room for improvements in our methods. We have added to the discussion of our paper some of the limitations of our work here using the Kinect V2 and our methods for estimating participant-specific BSPs

In addition to this, we believe that the open-source nature of our work will benefit those in the community interested in these methods. We provide full transparency and hope that the community can benefit from the extensive engineering work we have done. 

“Our work progresses on currently available tools, but further improvements need be considered before implementation. Our 3D scanning approach has the advantage that we directly measure the 3D shape, unlike regression modelling, other commonly used methods, or the elliptical cylinder method which uses 2D images to infer 3D shapes (4,25,30,56). By working directly in 3D our approach has the added advantage that assumptions about geometry can be minimized. Although we choose to use uniform density values, working in 3D also has the added advantage that non-uniform density functions can be implemented into the workflow, speaking to the flexibility of a surface scanning approach with programmatic and easily modifiable inputs (58). Indeed, we also found that our 3D scanning system had a measurement bias in underestimating total body mass for males (bias = -1.1 kg (-1.5%)) and females (bias = -3.1 kg (-4.4%)) when compared to the medical scale. A similar order of magnitude bias was reported for the use of older Kinect models for estimating volume and length (35,41). In one study, the trunk BSPs estimated using a geometrical model and several other approaches were compared to gold standard DEXA. There the authors found mean differences between the gold standard and the other approaches for trunk mass, center of mass, and moments of inertia ranging from overestimating by 18.3±15.1% to underestimating by -30.2±7.1% (57). Although the existence of a bias in our approach cannot be discounted and should be carefully considered, the range of this measurement error appears to be less than the possible errors from other widely used methods. By making our work open source we strive to provide the community with these tools and facilitate its use for further development to reduce bias and improve accuracy.” 

P.5 Additional detailed comments are not provided due to this reviewer’s belief that the sample size precludes this research from being a valuable contribution to the literature. With that said, if the same analytical procedures could be repeated in a much larger sample, I think this research would make a valuable contribution.

Response: We thank the reviewer for taking the time to provide us with helpful comments that have improved our manuscript. We have agreed with the reviewer that the sample size was small and have added appropriate changes to the discussion to clearly state this limitation. We hope the reviewer considers our added changes to the manuscript, our open-source work, and our clear limitations stated in the paper and see value in our contribution.

---

## [Decision Letter · Decision Letter 1]

21 Dec 2021

Estimating body segment parameters from three-dimensional human body scans

PONE-D-21-21310R1

Dear Dr. Kudzia,

We’re pleased to inform you that your manuscript has been judged scientifically suitable for publication and will be formally accepted for publication once it meets all outstanding technical requirements.

Kind regards,

Jeremy P Loenneke

Academic Editor

PLOS ONE

Additional Editor Comments (optional):

Reviewers' comments:

Reviewer's Responses to Questions

**Comments to the Author**

1. If the authors have adequately addressed your comments raised in a previous round of review and you feel that this manuscript is now acceptable for publication, you may indicate that here to bypass the “Comments to the Author” section, enter your conflict of interest statement in the “Confidential to Editor” section, and submit your "Accept" recommendation.

Reviewer #1: All comments have been addressed

Reviewer #2: All comments have been addressed

2. Is the manuscript technically sound, and do the data support the conclusions?

Reviewer #1: Yes

Reviewer #2: Yes

3. Has the statistical analysis been performed appropriately and rigorously? 

Reviewer #1: Yes

Reviewer #2: Yes

4. Have the authors made all data underlying the findings in their manuscript fully available?

Reviewer #1: Yes

Reviewer #2: Yes

5. Is the manuscript presented in an intelligible fashion and written in standard English?

Reviewer #1: Yes

Reviewer #2: Yes

6. Review Comments to the Author

Reviewer #1: (No Response)

Reviewer #2: (No Response)

7. PLOS authors have the option to publish the peer review history of their article (what does this mean?). If published, this will include your full peer review and any attached files.

Reviewer #1: No

Reviewer #2: No

---

## [Editor Report · Acceptance letter]

27 Dec 2021

PONE-D-21-21310R1 

Estimating body segment parameters from three-dimensional human body scans 

Dear Dr. Kudzia:

I'm pleased to inform you that your manuscript has been deemed suitable for publication in PLOS ONE. Congratulations! Your manuscript is now with our production department. 

Kind regards, 

on behalf of

Dr. Jeremy P Loenneke 

Academic Editor

PLOS ONE